# Genome-wide association studies and Mendelian randomization analyses for leisure sedentary behaviours

Yordi J. van de Vegte[1], M. Abdullah Said [1], Michiel Rienstra [1], Pim van der Harst [1,2,3,4✉] & Niek Verweij [1,5✉]

Leisure sedentary behaviours are associated with increased risk of cardiovascular disease, but whether this relationship is causal is unknown. The aim of this study is to identify genetic determinants associated with leisure sedentary behaviours and to estimate the potential causal effect on coronary artery disease (CAD). Genome wide association analyses of leisure television watching, leisure computer use and driving behaviour in the UK Biobank identify 145, 36 and 4 genetic loci ($P < 1 \times 10^{-8}$), respectively. High genetic correlations are observed between sedentary behaviours and neurological traits, including education and body mass index (BMI). Two-sample Mendelian randomization (MR) analysis estimates a causal effect between 1.5 hour increase in television watching and CAD (OR 1.44, 95%CI 1.25–1.66, $P = 5.63 \times 10^{-07}$), that is partially independent of education and BMI in multivariable MR analyses. This study finds independent observational and genetic support for the hypothesis that increased sedentary behaviour by leisure television watching is a risk factor for CAD.

[1] Department of Cardiology, University of Groningen, University Medical Center Groningen, 9700 RB Groningen, The Netherlands. [2] Department of Genetics, University of Groningen, University Medical Center Groningen, 9700 RB Groningen, The Netherlands. [3] Durrer Center for Cardiogenetic Research, Netherlands Heart Institute, 3511GC Utrecht, The Netherlands. [4] Department of Cardiology, University Medical Center Utrecht, 3584 CX Utrecht, The Netherlands. [5] Genomics plc, OX1 1JD Oxford, UK. ✉email: p.van.der.harst@umcg.nl; n.verweij@umcg.nl

Sedentary behaviours are defined as any waking behaviour characterized by an energy expenditure ≤1.5 metabolic equivalents, while in a sitting, reclining or lying posture[1]. In the United Kingdom, each adult spends an average 5 h sedentary per day[2]. Observational studies have previously shown that prolonged time spent on sedentary behaviours are associated with increased risk of cardiovascular disease and all-cause mortality[3–6]. Sedentary behaviours represent a major public health problem considering their high prevalence[2]. In addition, they pose a large economic burden on national level, with an estimated cost of cardiovascular disease caused by a sedentary lifestyle to be 424 million pounds sterling annually in the United Kingdom alone[7]. The association between sedentary behaviours and coronary artery disease (CAD) is less clear. Three studies that investigated the association between total sedentary behaviour and CAD found contrasting results[8–10], whereas one study focused on domain-specific sedentary behaviour and found a clear association between television watching and CAD[11].

In line with the previous literature on the association between sedentary behaviours and all-cause[3,5], cancer[3,5] and cardiovascular mortality[3,5], television watching seems most strongly associated with CAD. In fact, television watching is often used as proxy for total leisure sedentary behaviour in observational studies, as television watching is almost solely performed in occupational setting, modifiable by intervention[12–14] and shows higher validity than total sedentary behaviour as it is easier to recall[15]. Several hypotheses can be put forward as to why television watching seems to be the most important sedentary behaviour associated with risk of CAD. An epidemiological explanation is that leisure television time is associated with less and shorter breaks, lower total energy expenditure and different snacking behaviours than other sedentary traits, possibly increasing adverse effects of prolonged sitting[16–19]. A statistical explanation is that observational studies are possibly hampered by reverse causation or confounding through a broad range of determinants known to affect sedentary behaviours[20]. One recent study examined the causal nature of the association between leisure television watching and CAD by using a negative control outcome and found the association mirrored by the association with accidental death, indicating confounding to likely be a driver of the association between sedentary behaviours and CAD[21].

Another way to study causality of an association is by using a Mendelian randomization (MR) approach, which uses genetic variants as proxy for risk factors to minimize confounding and reversed causation in observational data[22,23]. Genetic variants are randomly assigned when passed from parents to offspring and are therefore mostly unrelated to the presence of confounders. However, little is known regarding the genetic variants accounting for the heritable component to domain-specific sedentary behaviours[24–26].

To extend our knowledge of how genetics might affect sedentariness and to investigate whether sedentary behaviours are a causal risk factor for CAD, we perform a genome-wide association study (GWAS) of three phenotypes of sedentary behaviours: (1) leisure television watching, (2) leisure computer use and (3) driving and identify 145, 36 and 4 genetic loci ($P < 1 \times 10^{-8}$), respectively. We find observational evidence that increased time spent watching television is associated with risk of CAD and find evidence for a similar causal estimate by using a MR approach. Considering the complex nature of behavioural traits and the broad range of determinants known to affect sedentary behaviours, we estimate the genetic correlation with other traits and find especially high correlations with educational and obesity traits. However, multivariable MR shows that the effect of television watching on CAD is at least partially independent of education and likely mediated by traditional cardiovascular risk

**Table 1 Baseline characteristics.**

|  | Sample | No CAD event | New-onset CAD |
|---|---|---|---|
| No. | 422,218 | 391,994 | 12,555 |
| Age, years | 57.4 ± 8.0 | 57.1 ± 8.0 | 61.6 ± 6.4 |
| Sex, male | 45.7 | 43.7 | 66.4 |
| BMI, kg m$^{-2}$ | 27.4 ± 4.7 | 27.3 ± 4.7 | 28.8 ± 4.9 |
| Diabetes mellitus, % | 5.1 | 4.2 | 11.8 |
| Essential hypertension, % | 8.2 | 6.2 | 17.8 |
| Smoking behaviour, % | | | |
| Ideal | 54.7 | 56.0 | 41.7 |
| Intermediate | 35.0 | 34.0 | 41.9 |
| Poor | 10.3 | 10.0 | 16.4 |
| Physical activity behaviour, % | | | |
| Ideal | 66.6% | 67.2% | 60.3% |
| Intermediate | 24.7% | 24.6% | 25.9% |
| Poor | 8.6% | 8.2% | 13.8% |
| Alcohol use, 10 ml per week | 16.6 ± 17.0 | 16.5 ± 16.8 | 18.7 ± 19.7 |
| Townsend deprivation index | −0.07 ± 0.97 | −0.08 ± 0.97 | 0.05 ± 1.01 |
| Maximum years spent on education | 14.6 ± 4.8 | 14.7 ± 4.8 | 13.2 ± 5.1 |
| Television watching, hours per day | 2.8 ± 1.5 | 2.8 ± 1.5 | 3.2 ± 1.7 |
| Computer use, hours per day | 1.0 ± 1.2 | 1.0 ± 1.2 | 1.0 ± 1.3 |
| Driving, hours per day | 0.9 ± 1.0 | 0.9 ± 1.0 | 1.0 ± 1.1 |

Continuous variables are presented as mean ± SD and binary variables as percentages.
BMI body mass index, CAD coronary artery disease.

factors as body mass index (BMI). These results support conclusions from traditional observational epidemiology that policy guidelines aiming to reduce sedentary behaviours may help to prevent CAD.

**Results**

**Baseline characteristics.** Of 501,105 individuals who responded to at least one of the three leisure sedentary time questions, 81 were excluded for analyses on a per-phenotype basis, 1337 failed genetic quality control and 77,469 were of non-European ancestry. In total, 422,218 individuals of European ancestry from the UK Biobank were included in this study (Table 1). 45.7% of the participants were male, and the average age of the cohort was 57.4 (SD 8.0) years old at the time of first assessment. Mean daily reported leisure television watching was 2.8 h (SD 1.5), leisure computer use was 1.0 h (SD 1.2) and driving was 0.9 h (SD 1.0).

**Association of sedentary behaviour traits.** First, we examined the correlations among the three sedentary phenotypes in the UK Biobank. Leisure television watching was inversely correlated with computer use ($r = -0.05$, $P = 1.1 \times 10^{-230}$) and driving ($r = -0.03$, $P = 4.8 \times 10^{-57}$), while leisure computer use and driving had a weak positive correlation ($r = 0.05$, $P = 1.91 \times 10^{-232}$) (Fig. 1a, Supplementary Table 1). Associations with possible confounding risk factors are shown in Supplementary Table 2.

**Association of sedentary behaviour traits with CAD.** Next, we examined the association of baseline sedentary behaviour traits with risk of incident CAD using Cox proportional hazard regression analyses. During an average median follow-up time 6.3 years in the UK Biobank, 12,593 individuals developed new-onset CAD. Leisure television watching and driving were significantly associated with CAD in the univariate analyses (respectively, HR 1.20, 95% CI 1.18–1.21, $P = 2.7 \times 10^{-233}$; HR 1.03, 95% CI 1.01–1.04, $P = 5.6 \times 10^{-4}$), while computer use was not (Fig. 1b, Supplementary Table 3). Only leisure television watching remained significantly associated with CAD events (HR 1.03, 95% CI 1.02–1.04, $P = 1.2 \times 10^{-6}$) after extensive adjustment for cardiovascular risk factors (Fig. 1b, Supplementary Table 3).

**Genome-wide analyses of sedentary behaviours.** A GWAS of sedentary behaviours was performed in 408,815 UK Biobank subjects of European descent, using 19,400,418 directly genotyped

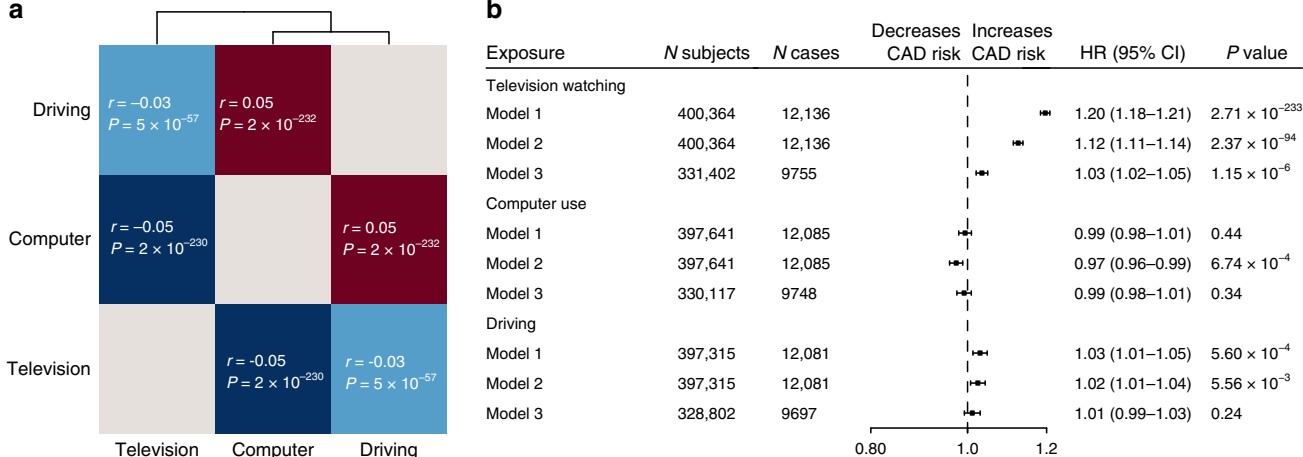

**Fig. 1 Results of the observational statistical analyses performed on sedentary behaviours. a** Heatmap of *z*-scores for the associations between sedentary phenotypes. Numbers in white show the correlations. **b** Forestplot depicting the results of the two-sided Cox regression analysis between sedentary phenotypes and CAD. On the *X*-axis, hazard ratios are shown and results are represented as hazard ratio and 95% confidence intervals. Three regression models were used to study the association between sedentary behaviour phenotypes and CAD events; Model 1: univariable analysis; Model 2: adjusted for age and sex; Model 3: adjusted for age, sex, body mass index, smoking status, hypertension, diabetes, Townsend deprivation index as proxy for income, physical activity levels, alcohol use per weak and years of education. We considered two-sided *P* < 0.05 statistically significant, no adjustments were made for multiple testing. R correlation, HR hazard ratio, CI confidence interval.

and imputed autosomal genetic variants. Three phenotypes of sedentary behaviour were studied: leisure television watching, leisure computer use and driving. The GWAS revealed 193 variants in 169 loci associated with one or more sedentary traits (Fig. 2a, Supplementary Data 1). The large majority, 152 independent variants in 145 loci, were associated with leisure television watching (Supplementary Data 1, Supplementary Fig. 1). In addition, we found 37 independent variants in 36 loci for leisure computer use (Supplementary Data 1, Supplementary Fig. 2) and four independent variants in four loci for spent driving (Supplementary Data 1, Supplementary Fig. 3). Television watching and computer use showed some overlap with 15 shared loci (Fig. 2b). Interestingly, 8 out of these 15 loci had opposing effects between television watching and computer use (Supplementary Data 1). The same was true for the shared locus between television watching and driving (Supplementary Data 1). SNP-heritability as estimated by BOLT-REML was highest for television watching ($h^2_g = 0.161$, se = 0.002), followed by leisure computer use ($h^2_g = 0.093$, se = 0.002), and driving ($h^2_g = 0.044$, se = 0.002). Television watching and computer use showed a negative genetic correlation ($r_g = -0.281$, se = 0.011, $P = 6.17 \times 10^{-144}$), similar to the observational analyses. Driving was positively associated with television watching ($r_g = 0.231$, se = 0.016, $P = 3.00 \times 10^{-47}$), but not with computer use ($r_g = 0.013$, se = 0.019, $P = 0.494$). Television watching and computer use showed a positive genetic correlation with objectively measured sedentary behaviour (respectively, $r_g = 0.145$, se = 0.0284, $P = 2.97 \times 10^{-7}$; $r_g = 0.4571$, se = 0.03, $P = 4.23 \times 10^{-52}$), while driving did not ($r_g = -0.029$, se = 0.047, $P = 0.535$). Genetic correlations with other traits can be found in Supplementary Data 2. The GWAS Catalog was queried to find previously established genetic variants in LD ($R^2 > 0.1$) with the newly discovered variants. Of the 193 variants, 21 genetic variants were in linkage disequilibrium ($R^2 > 0.8$) with previously established variants for years of education and 46 variants with any trait surpassing $P < 1 \times 10^{-5}$ (Supplementary Data 3).

**Candidate genes and insights into biology**. We explored the potential biology of the 169 identified loci by prioritizing candidate causal genes in these loci (Supplementary Data 1): 185

unique genes were in proximity (the nearest gene and any additional gene within 10 kb, Supplementary Data 1) of the lead variant, 19 unique coding variants in LD with sedentary behaviour variants (Supplementary Data 4), 27 unique genes were selected based on multiple functional expression quantitative trait loci (eQTL) analyses (Supplementary Data 5), and 75 unique genes were prioritized based on DEPICT analyses (Supplementary Data 6). Of the 306 candidate causal genes identified, 56 genes were prioritized by multiple methods of identification, which may be used to prioritize candidate causal genes (Fig. 2c).

**Pathway analyses and tissue enrichment**. Pathway analysis was performed for all sedentary traits combined and for each trait separately (Supplementary Data 7). All pathways revolved around common themes including neurological development, neuronal longevity and signalling pathways. Importantly, 361 reconstituted gene sets were found to be associated with television watching, compared with none for computer use. In total, 41 of the 143 suggestively associated (false discovery rate < 0.20) gene sets of computer use were associated with television watching as well (false discovery rate < 0.05). The tissue enrichment analyses by DEPICT implicated the nervous system as the most important tissue with 22 of the 24 enriched tissues located within the nervous system (Supplementary Data 8). No pathways or tissues were highlighted for driving (Supplementary Data 7, 8).

**Causal relationship between sedentary behaviours and CAD**. A series of MR analyses was performed to test the hypothesis that increased television watching, computer use and driving are causal risk factors for CAD. Results of these analyses are shown in Fig. 3, Supplementary Table 4, Supplementary Figs 4–6 and are discussed below.

Using the IVW-MR fixed-effects approach, a causal effect was estimated between a 1 SD increase in leisure time watching television and CAD (OR 1.44, 95% CI 1.28–1.62, $P = 7.28 \times 10^{-10}$), as well as between driving and CAD (OR 2.65, 95% CI, 1.35–5.19, $P = 4.46 \times 10^{-3}$). However, this was not true for leisure computer use (OR 0.81, 95% CI 0.64–1.02, $P = 0.07$). Television watching remained significantly associated with CAD in all univariable pleiotropy and sensitivity analyses, except

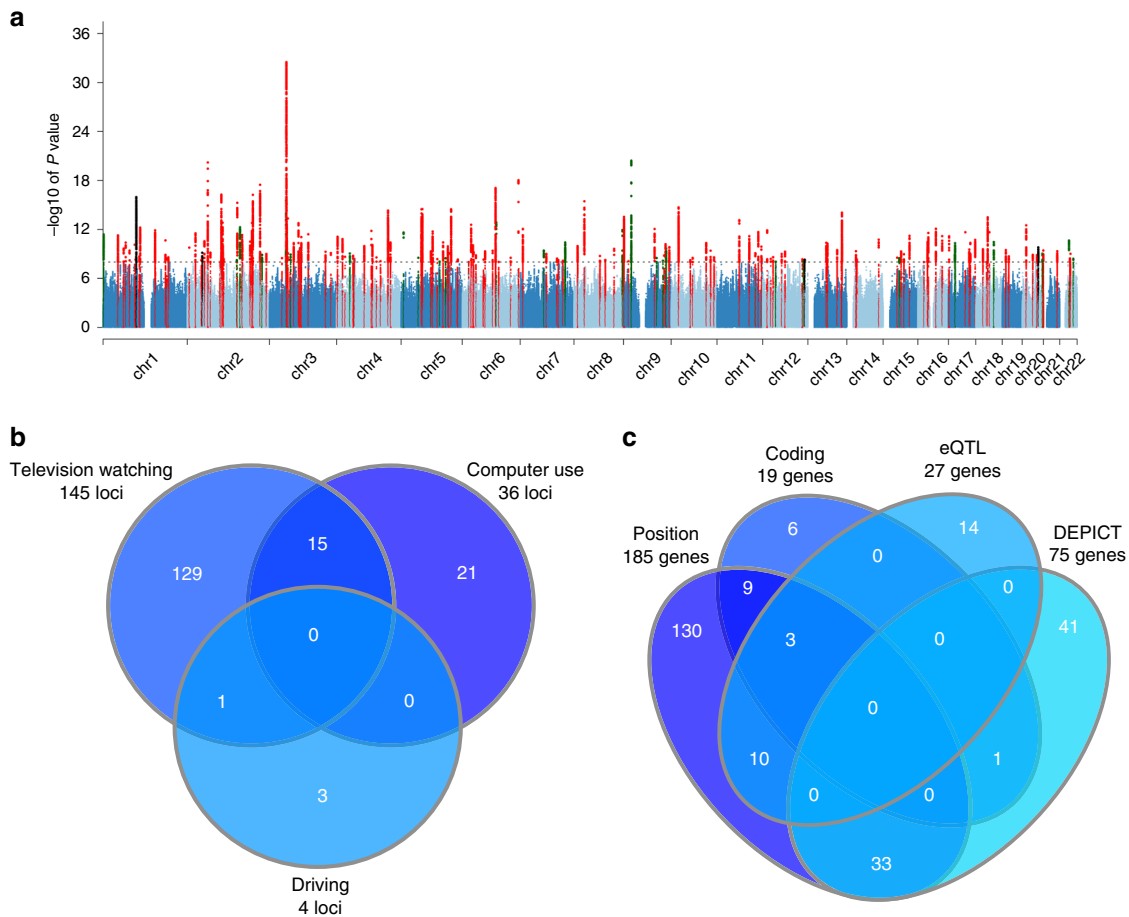

**Fig. 2 Results of genome-wide and candidate gene analyses of sedentary behaviours. a** Overlay Manhattan plot shows the results for the genome-wide associations with sedentary behaviour traits among individuals of European origin. Loci reaching genome-wide significance ($P < 1 \times 10^{-8}$) are coloured red for television watching, green for computer use and black for driving. **b** Venn plot shows overlap between loci found for different sedentary behaviours. **c** Venn plot shows overlap of genes tagged by one or multiple strategies. EQTL expression quantitative trait loci, DEPICT data-driven expression prioritized integration for complex traits.

weighted mode, which had wider confidence intervals but a similar effect estimate (Fig. 3a, Supplementary Table 4, Supplementary Fig. 4). Computer use, in line with the main analyses, was not significantly associated with CAD (Fig. 3b, Supplementary Table 4, Supplementary Fig. 5). The relationship between driving and CAD was not consistent across pleiotropy analyses (Fig. 3c, Supplementary Table 4, Supplementary Fig. 6). We lowered the $P$ value thresholds considering the low number of variants used in the MR of driving. The genetic association was not robust to $P$ value thresholds, since not only the confidence intervals were larger for several sensitivity analyses, but estimates were reversed as well (Supplementary Table 4). Not a single variant was removed due to MR-Steiger filtering and results remained unchanged.

Instruments used for the two-sample univariable MR analyses between sedentary behaviours and CAD can be found in Supplementary Data 9. The instruments for the main analyses ($P < 1 \times 10^{-8}$) had F-statistics ranging from 29 to 164 for television watching, from 31 to 83 for computer use and from 35 to 46 for driving (Supplementary Data 10), suggesting low chance of weak instrument bias. $I^2_{GX}$ of 0.98 for television watching and 0.98 for computer use indicated low chance of weak instrument bias in the MR-Egger analyses (Supplementary Table 5). $I^2_{GX}$ for driving was 0, suggesting a high chance of weak instrument bias. All excluded variants in the MR pleiotropy residual sum and outlier (MR-PRESSO) analyses are shown in

Supplementary Table 6. Excluded genetic variants with potential pleiotropic effects through education and through all traits are summarized in Supplementary Table 7.

In order to explore pleiotropy, we investigated heterogeneity using the $I^2$ index and Cochran's Q for MR-IVW analyses and Rucker's Q for MR-Egger analyses (Supplementary Table 5). Using Cochran's Q, heterogeneity and thus potential balanced pleiotropy was found for leisure television watching. We therefore used the MR-IVW random-effects approach to assess the association between television watching and CAD, which remained significant (OR 1.44, 95% CI 1.25–1.66, $P = 5.63 \times 10^{-07}$). Rucker's Q was not significantly lower than Cochran's Q for television watching, computer use and driving (Supplementary Table 5), indicating absence of unbalanced horizontal pleiotropy and thus suggesting MR-IVW analyses to be the best approach. Additional forestplots to visually inspect heterogeneity are provided in Supplementary Figs. 7–9. MR-Egger intercepts $P$ values were >0.05 (Supplementary Table 5), which suggests an absence of bias due to pleiotropy in the MR-IVW analyses.

Using the multivariable MR approach correcting for education, we found the direct effect of television watching on CAD to be attenuated compared with the total effect due to a wider confidence interval. However, a causal effect between 1 SD increase in leisure time watching television was still estimated to increase CAD risk when corrected for education (OR 1.42, 95% CI 1.09–1.84, $P = 8.70 \times 10^{-03}$). Please see Supplementary

**a** Television watching - coronary artery disease

| Method | N snps | Decreases CAD risk / Increases CAD risk | OR (95% CI) | P value |
|---|---|---|---|---|
| Inverse variance weighted (fixed effects) | 127 | | 1.44 (1.28–1.62) | $7.28 \times 10^{-10}$ |
| Inverse variance weighted (random effects) | 127 | | 1.44 (1.25–1.66) | $5.63 \times 10^{-07}$ |
| MR-Egger | 127 | | 2.16 (1.12–4.16) | 0.02 |
| Inverse variance weighted (excluding pleiotropic education SNPs) | 113 | | 1.44 (1.27–1.63) | $1.93 \times 10^{-08}$ |
| Inverse variance weighted (excluding pleiotropic SNPs) | 93 | | 1.50 (1.31–1.73) | $1.26 \times 10^{-8}$ |
| MR–PRESSO | 127 | | 1.44 (1.25–1.66) | $1.85 \times 10^{-06}$ |
| MR–PRESSO (outlier–corrected) | 126 | | 1.47 (1.28–1.68) | $2.56 \times 10^{-07}$ |
| Multivariable (adjusted for education) | 126 | | 1.42 (1.09–1.84) | $8.70 \times 10^{-03}$ |
| Weighted median | 127 | | 1.44 (1.20–1.72) | $7.82 \times 10^{-05}$ |
| Weighted mode | 127 | | 1.47 (0.90–2.41) | 0.13 |

0.5   1.0   1.5   2.0   2.5   3.0   3.5

**b** Computer use - coronary artery disease

| Method | N snps | Decreases CAD risk / Increases CAD risk | OR (95% CI) | P value |
|---|---|---|---|---|
| Inverse variance weighted (fixed effects) | 32 | | 0.81 (0.64–1.02) | 0.07 |
| Inverse variance weighted (random effects) | 32 | | 0.81 (0.60–1.09) | 0.17 |
| MR-Egger | 32 | | 0.20 (0.02–2.36) | 0.21 |
| Inverse variance weighted (excluding pleiotropic education SNPs) | 28 | | 0.89 (0.69–1.15) | 0.37 |
| Inverse variance weighted (excluding pleiotropic SNPs) | 27 | | 0.89 (0.69–1.15) | 0.38 |
| MR–PRESSO | 32 | | 0.81 (0.60–1.09) | 0.18 |
| MR–PRESSO (outlier–corrected) | 31 | | 0.87 (0.67–1.14) | 0.33 |
| Multivariable (adjusted for education) | 32 | | 1.17 (0.71–1.92) | 0.53 |
| Weighted median | 32 | | 0.82 (0.58–1.14) | 0.24 |
| Weighted mode | 32 | | 0.80 (0.43–1.49) | 0.49 |

0.5   1.0   1.5   2.0   2.5   3.0   3.5

**c** Driving - coronary artery disease

| Method | N snps | Decreases CAD risk / Increases CAD risk | OR (95% CI) | P value |
|---|---|---|---|---|
| Inverse variance weighted (fixed effects) | 4 | | 2.65 (1.35–5.19) | $4.46 \times 10^{-03}$ |
| Inverse variance weighted (random effects) | 4 | | 2.65 (1.35–5.19) | $4.46 \times 10^{-03}$ |
| MR-Egger | 4 | | 0.49 (0.002–129.32) | 0.82 |
| Inverse variance weighted (excluding pleiotropic education SNPs) | 3 | | 2.87 (1.30–6.30) | $8.86 \times 10^{-03}$ |
| Inverse variance weighted (excluding pleiotropic SNPs) | 2 | | 1.99 (0.78–5.10) | 0.15 |
| MR–PRESSO | 4 | | 2.65 (1.52–4.63) | 0.04 |
| MR–PRESSO (outlier–corrected) | NA | | NA | NA |
| Multivariable (adjusted for education) | 4 | | 3.31 (1.31–8.38) | 0.01 |
| Weighted median | 4 | | 2.10 (0.95–4.67) | 0.07 |
| Weighted mode | 4 | | 2.07 (0.72–5.98) | 0.27 |

0.5   1.0   1.5   2.0   2.5   3.0   3.5

Tables 8–10 and Supplementary Data 11 for the results of the MR between educational years and sedentary behaviours and Supplementary Table 4 for the multivariable MR between sedentary behaviours, educational years and CAD. We found no evidence for weak instrument bias within the multivariable MR setting, but $Q_a$ indicated remaining heterogeneity and thus potential pleiotropy in the estimates between television watching, education and CAD (Supplementary Table 11). The direct effect of television watching on CAD was attenuated compared with the total effect when corrected for BMI (OR 1.28, 95% CI 1.05–1.55,

**Fig. 3 Summary Mendelian randomization (MR) estimates of leisure sedentary behaviours on CAD.** Summary MR estimates of the causal association between (**a**) leisure television watching, (**b**) leisure computer use and (**c**) driving on coronary artery disease were derived from the main inverse-variance-weighted (MR-IVW), MR-Egger, MR-IVW excluding potentially pleiotropic single-nucleotide polymorphisms (SNPs) through education, MR-IVW excluding potentially pleiotropic SNPs through any trait, MR pleiotropy residual sum and outlier (MR-PRESSO), outlier-corrected MR-PRESSO, multivariable Mendelian randomization adjusted for educational years, weighted median and weighted mode-based estimator methods. On the X-axis, odds ratios are shown and data are represented as odds ratio and 95% confidence intervals. We considered two-sided $P < 0.05$ statistically significant, no adjustments were made for multiple testing. OR odds ratio, CI confidence interval.

$P = 0.01$), low density lipid protein levels (OR 1.44, 95% CI 1.25–1.66, $P = 6.2 \times 10^{-07}$), a history of diabetes (OR 1.31, 95% CI 1.11–1.55, $P = 1.64 \times 10^{-03}$) and hypertension (OR 1.25, 95% CI 1.06–1.48, $P = 8.2 \times 10^{-03}$). Please see Supplementary Table 12 for the full results. Instruments used for the two-sample multivariable MR between sedentary behaviours and CAD when corrected for education and when corrected for other cardiovascular risk factors, can be found in Supplementary Data 12, 13, respectively.

## Discussion

We report the identification of 169 loci GWAS loci for sedentary behaviours. Our observational and genetic analyses provide complementary evidence that leisure television watching is causally associated with CAD.

We discovered 169 distinct genetic loci for sedentary behaviours, of which 16 loci showed an overlap between two sedentary behaviour traits. Interestingly, 9 out of the 16 loci had opposing effects between television watching and computer use. Observational and genetic correlations between sedentary behaviour traits were weak and similar between both approaches for television watching and computer use. Genetic pathways revolved around neurological theme for leisure television watching and computer use, although different pathways were implicated. We did not find any pathways suggestively associated with driving presumably due to limited power. The important role of the central nervous system is analogous to earlier GWAS on physical activity questionnaires and device-measured activity data[27,28]. We found the highest genetic correlations between sedentary behaviours and educational traits, which were negative for television watching and driving, and positive for computer use. In addition, sedentary behaviours as measured by leisure television watching and driving were correlated with obesity traits, but not waist–hip ratio. This is in accordance with the current understanding that overall fat distribution is mainly neurologically driven[29], but waist–hip ratio by adipose pathways[30]. Intriguingly, computer use was not correlated with any obesity trait. This suggests that questionnaires have the ability to capture domain-specific aspects of sedentary behaviour as previously described[15] and possibly share complex genetic patterns with education-related traits and cardio-metabolic risk factors.

Only three candidate gene studies (investigating *FTO*, *DRD2* and *MC4R*) have been performed to understand the biology of domain-specific sedentary behaviours[31–33]. However, none of these genes or variants were brought forward in our analysis. One recent GWAS on blood pressure[34] performed a phenome scan and reported locus rs13107325 to be related to leisure television watching, which is confirmed in the current study.

The genetic correlation between television watching and objectively measured sedentary behaviour was weak, in accordance to previous findings from observational studies[35]. This is at least in part because accelerometers can only measure total sedentary time and not domain-specific behaviours, such as television watching[15]. The correlation between computer use and objectively measured sedentary behaviour was higher; one possible explanation for the current finding is a volunteering bias in the accelerometer data, which could have put forward individuals who were more highly educated and spent more leisure time on the computer.

Our data provides observational evidence in support of the hypothesis that sedentary behaviour by leisure television watching is a risk factor for CAD. This association could not be established for leisure computer use and driving. The association between television watching and CAD was significant in the main MR analyses and consistent across follow-up uni-variable MR sensitivity analyses. The multivariable MR in which we corrected for education indicated horizontal pleiotropy due to education. Although confidence intervals were broader, the results showed an effect of television watching on CAD independent of education. The multivariable MR analyses correcting for traditional cardiovascular risk factors indicated vertical pleiotropy, as the direct effects of television watching on CAD were attenuated compared with the total effects. This provides genetic insights in how complex traits as sedentary behaviours are associated with CAD. For leisure computer use, we could not identify a meaningful association with CAD, in line with the observational analyses. In contrast to the observational analyses, we found an association between driving and CAD using a MR approach. However, this relationship could not be established in several sensitivity analysis as it seemed to be caused by potentially pleiotropic variants. Therefore, we are cautious to determine driving as a causal risk factor for CAD.

The key strength of the current study is that we are the first to combine a observational and genetic approach to assess the association between domain-specific sedentary behaviours and CAD, as triangulation of evidence strengthens the conclusion that the effect is causal[36]. Furthermore, we here report on the genetics explaining inter-individual differences in domain-specific leisure sedentary behaviours and subsequently used these genetic variants as instruments in MR analyses to potentially overcome confounding as source of bias that plague observational studies.

This study also has limitations. First of all, the genetic instruments could be non-specific to sedentary behaviours, as a statistical and not a biological approach was used for their selection. Several other sources of bias may be at play in the MR that should be acknowledged to correctly interpret the results. To address the assumptions of MR and control for different types of biases, we performed several sensitivity analyses according to the latest guidelines[37], each with their own strengths and weaknesses that are more extensively described in the Supplementary Discussion. It is important to recognize these limitations and strengths in light of the potential complicated relationship between education and disease[38]. Furthermore, we found evidence for heterogeneity and thus potential pleiotropy in the multivariable MR corrected for education, suggesting that unobserved confounders could play a role in the relationship between television watching and CAD. As far as verifiable using currently available methods[37], all results point to the same direction and therefore seem to support the rationale that interventions targeting television watching may reduce CAD risk[12–14], especially considering the high prevalence[2] and non-occupational characteristics of television watching. We

advocate that the data presented here should be re-analysed when MR methods to account for pleiotropy are further developed.

Currently, the ability to replicate genetic variants in external cohorts is limited due a lack of available data concerning the same sedentary behaviour questions and genetics. We therefore adopted stringent thresholds for genome-wide significance. We also note the limited generalizability to individuals of non-European ancestry. Both the observational and MR study are limited by quality of the questionnaires and the effectiveness of measurements to capture features that are on the causal pathways. However, MR studies are less likely to be affected by measurement error on the exposures than conventional observational analyses[39]. The data used to obtain information about sedentary behaviours were subjectively measured, which are known to underestimate the actual sedentary time[40], any possible measurement errors are likely biased towards the null. In addition, since the questionnaire did not include occupational sedentary behaviours, conclusions cannot be generalized to total sedentary behaviour. Future research efforts should be directed at expanding the current set of analyses to total sedentary behaviour, physical activity and sleep behaviours, including accelerometer data, when new cohorts with sufficient genetic data become available. Finally, the current analyses were performed using data of individuals aged between 40 and 69. Of all age groups, this is the group that spends most time watching television[41]. Environmental differences, such as changes in television viewing habits of younger people, could affect the estimated effects; extrapolation of the current findings to a younger population should be subjected to further research.

In conclusion, we provide evidence for a causal estimate between sedentary behaviour as measured by television watching and CAD. However, the results also indicate that there are uncertainties in these estimates due to potential horizontal pleiotropy by education, which are difficult to entangle using current state-of-the-art data and MR techniques. These results support conclusions from traditional observational epidemiology that policy guidelines aiming to reduce sedentary behaviours may prevent CAD.

## Method
**Study population**. The UK Biobank is a large, population-based cohort consisting of 503,325 individuals aged 40–69 years that were included by general practitioners of the UK National Health Service (NHS) between 2006 and 2010. All study participants provided informed consent and the North West Multi-centre Research Ethics Committee approved the study[42]. Detailed methods used by UK Biobank have been described elsewhere[43].

**Ascertainment of sedentary time**. During the first visit, participants were asked three questions, "In a typical DAY, how many hours do you spend watching TV?", "In a typical DAY, how many hours do you spend using the computer? (Do not include using a computer at work)" and "In a typical DAY, how many hours do you spend driving?". Participants outside a 99.5% range on the right side of the normal distribution were excluded on a per-phenotype basis, since the sedentary phenotypes were right-skewed.

**Observational statistical analyses**. The associations between sedentary phenotypes were assessed by performing Spearman's rank correlation. Cox regression analysis was performed to investigate the association between different sedentary behaviours and new-onset CAD events. CAD was defined based on ICD-9 and ICD-10 codes, together with operation codes and self-reported data on myocardial infarction, other ischaemic heart disease and history of coronary artery bypass grafting or percutaneous coronary intervention as performed previously[44]. Three Cox regression models were used to investigate the association of leisure sedentary television watching, leisure computer use and driving with CAD; potential confounders were selected per prior epidemiological analyses[8]. These models included (1) univariable analysis (2) a multivariable model correcting for age and sex (3) a multivariable model correcting for important CAD risk factors. These risk factors included age, sex, BMI, smoking status, hypertension, diabetes, Townsend deprivation index as proxy for income, physical activity levels, alcohol use per weak and years of education. Associations between sedentary behaviours and potential confounders were assessed using linear regressions analyses or logistic regression

analyses in case of binary outcomes. We considered two-sided $P < 0.05$ statistically significant. Analyses were performed using statistical software STATA 15 (StataCorp LP).

**Covariate definitions**. The covariates age, BMI, Townsend deprivation index and alcohol use were treated as continuous variables. The Townsend deprivation index is a measure of material deprivation within a population based on unemployment, non-car ownership, non-home ownership and household overcrowding[45]. We single-inverse normalized the Townsend deprivation index, in line with previous studies' methodologies[46]. Sex, smoking status, hypertension, diabetes and years of education were ordinal data. Smoking status was defined as ideal (never smoked or quit >12 months ago), intermediate (quit smoking ≤12 months ago) or poor (current smoker). Physical activity was based on questionnaires concerning do-it-yourself and exercise activities using guidelines for ideal cardiovascular health and according to previous research[44]. In short, physical activity was defined as ideal if participants had ≥150 min/week moderate or ≥75 min/week vigorous or 150 min/week mixed (moderate and vigorous) activity. Intermediate physical activity was more than 1 min/week of moderate or vigorous exercise without achieving ideal physical activity guidelines. Poor physical activity was defined as not performing any moderate or vigorous activity. Duration and intensity of physical activity was ascertained using the answers provided by participants on a range of questions based on the validated International Physical Activity Questionnaire[47]. Do-it-yourself activity examples included pruning or lawn watering for light activities, and lifting heavy objects, using heavy tools or digging for heavy activities. Years of education were based on the standardized 1997 international standard classification of education (ISCED) according to previously published guidelines[48].

**Genotyping and imputation**. The Wellcome Trust Centre for Human Genetics performed genotyping, quality control before imputation and imputed to HRC v1.1 panel. Analysis has been restricted to variants that are in the HRC v1.1. Quality control of samples and variants, and imputation was performed by the Wellcome Trust Centre for Human Genetics, as described in more detail elsewhere[49]. Minor Allele Frequency of 0.5% and INFO-score of more than 0.3 was used in post-GWAS analysis.

**Genome-wide association study**. All three sedentary phenotypes were inverse rank normalized in order to obtain normally distributed data. Genome-wide association analysis in UK Biobank was performed using BOLT-LMM v2.3beta2, employing a mixed linear model that corrects for population structure and cryptic relatedness[50]. Leisure television watching, leisure computer use and driving were adjusted for age-squared, age, sex, age-sex interaction, the first 30 principal components to correct for population stratification and genotyping array (Affymetrix UK Biobank Axiom array or Affymetrix UK BiLEVE Axiom array). Participants were excluded if they were of non-European ancestries ($n = 78,372$) in order to reduce non-polygenetic signals.

We used the PLINK clumping procedure for each sedentary phenotype separately to prune genetic variants at a stringent linkage disequilibrium (LD) of $R^2 < 0.005$ within a five megabase window into a set of independently associated variants. Genetic loci were determined by assessing the highest associated variants in a one megabase region at either side of the independent variants. We combined all loci of the sedentary phenotypes and again searched within a one megabase region at either side to obtain the highest associated locus in order to receive a set of independent genetic loci associated with sedentary behaviour in general (Supplementary Data 1).

Since the current study is the only population-based study of sedentary behaviours, independent cohorts that matched this study in size and availability of variables (specific questions assessing different subtypes of sedentary behaviour combined with genetics) were unavailable for replication purposes. Therefore, only loci that reached a stringent (two-sided) genome-wide significant threshold of $P < 1 \times 10^{-8}$ were taken forward, in order to account for multiple independent traits in line with other multi phenotype studies[51,52].

The genomic inflation lambda was 1.37 for watching television, 1.20 for computer use and 1.10 for time spent driving. LD score regression intercepts showed no genomic inflation due to non-polygenic signals for computer use ($1.049 \pm 0.0092$) and time spent driving ($1.0083 \pm 0.0076$). Attenuation ratio statistic[53] indicated polygenicity, not population stratification, to be the main driver of the observed inflation of test statistics for television watching ($0.0697 \pm 0.0104$). QQ-plots for the three independent GWAS traits can be found in Supplementary Figs. 10–12.

**Functional annotation of genes and pathway analyses**. Genetic correlations between the three sedentary phenotypes were assessed using BOLT-REML variance components analysis using BOLT v2.3.1[54]. In addition, genetic correlations with other traits were assessed using LD score regression software (v1.0.0)[55,56] and the LD hub platform (v1.9.3)[57]. Genetic correlations were considered significant if they achieved a Bonferroni-corrected significance of $P < 0.05/696 = 7.18 \times 10^{-5}$.

**Functional annotation of genes and pathway analyses**. For all independent genetic variants that were genome-wide significantly associated with a sedentary

behaviour, candidate causal genes were prioritized as follows: (1) by proximity, the nearest gene or any gene within 10 kb; (2) genes containing coding variants in LD with sedentary variants at $R^2 > 0.8$; 3) eQTL genes in LD ($R^2 > 0.8$) with sedentary behaviour variants (described below); and (4) DEPICT gene mapping using variants that achieved $P < 1 \times 10^{-6}$ (described below).

**eQTL analyses.** To search for evidence of the functional effects of genetic variants associated with any of the three sedentary traits, we used multiple functional eQTL mapping. This was done using summary data based Mendelian randomization (SMR)[58] analysis (version 0.710) in data repositories from GTEx V7[59], GTEx brain[60], Brain-eMeta eQTL[60] and blood eQTL from Westra[61] and CAGE[62]. EQTL genes were considered as a candidate causal gene if they achieved a Bonferroni-corrected significance of $P < 0.05/187,747 = 2.66 \times 10^{-7}$, passed the HEIDI test of $P > 0.05$ and if the lead variants of the eQTL genes were in LD ($R^2 > 0.8$) with the queried variants.

**Pathway analyses by DEPICT.** Identification of genes associated with identified variants, enriched gene sets and tissues in which these genes are highly expressed, was performed using DEPICT. DEPICT.v1.beta version rel137 (obtained from https://data.broadinstitute.org/mpg/depict/) was used to perform integrated gene function analyses. DEPICT was run using all genetic variants that achieved $P < 1 \times 10^{-6}$. We opted for a more conservative approach than suggested because the signal was highly polygenic[63].

**MR analyses.** All 1000G imputed independent lead variants associated with sedentary behaviours at $P < 1 \times 10^{-8}$ were used as instrumental variables in the main MR using inverse-variance-weighted fixed-effects meta analyses. For all MR analyses, we considered two-sided $P < 0.05$ as statistically significant. MR analyses were performed using the R package TwoSampleMR (version 0.4.20), MR-PRESSO (version 1.0)[64] and MVMR (version 0.1)[65].

For CAD, the Coronary Artery Disease Genome-Wide Replication and Meta-Analysis plus Coronary Artery Disease Genetics Consortium's (CARDIoGRAMplusC4D)[66] 1000 genomes-based meta-analysis was used. This cohort included data from individuals of mostly European, but also Hispanic, African American, and South and East Asian ancestry. In total, there were 60,801 cases of CAD and 123,504 control subjects[66]. CAD events were defined as a documented diagnosis of CAD, such as acute coronary syndrome (including MI), chronic stable angina, or >50% stenosis of at least one coronary vessel, as well as those who had undergone percutaneous coronary revascularization or coronary artery bypass grafting[66].

For educational years, the Social Science Genetic Association Consortium (SSGAC) GWAS meta-analysis of years of schooling was used[48]. We used information of the cohort which included 293,723 individuals of European ancestry, as this cohort did not include the UK Biobank. Educational years were standardized using the 1997 ISCED of the United Nations Educational, Scientific and Cultural Organization[48]. Proxies were not searched for in case requested lead variants of sedentary behaviours were not found in the GWAS of CAD or educational years.

**Weak instrument bias in MR analyses.** The strength of the instruments was assessed using the $F$-statistic, calculated using the equation $F = R^2(n - 2)/(1 - R^2)$[67]. In this formula, $R^2$ is the proportion of the variability in sedentary behaviours explained by the SNP and $n$ is the sample size[67]. An F-statistic of >10 indicates a relatively low risk of weak instrument bias in MR analyses[67], which is essential prevent violation of the 'NO Measurement Error' assumption. In addition, potential weak instrument bias in MR-Egger regression analysis was assessed by calculating the variation between individual genetic variant estimates for each exposure ($I^2_{GX}$)[68]. An $I^2_{GX}$ of >95% was considered low risk of measurement error.

**Pleiotropy analyses in MR analyses.** Pleiotropy in the context of MR analyses refers to genetic variants exerting multiple effects. In other words, pleiotropic genetic variants may affect the outcome independently of the exposure. This can lead to confounding and bias of MR estimates and investigation of pleiotropy is therefore essential. First, $I^2$- index[69] and Cochran's Q[70] statistics were calculated to test for heterogeneity produced by different genetic variants in the fixed-effect variance weighted analyses. Heterogeneity statistics provide useful information on pleiotropy, since low heterogeneity indicates that estimates between genetic variants should vary by chance only, which is only possible in case of absence of pleiotropic effects. An $I^2$ index > 25% and Cochran's Q $P$ value of <0.05 were considered as an indication of heterogeneity and, as a consequence, of pleiotropy. In case Cochran's Q indicated potential pleiotropy, we moved from the inverse-variance-weighted fixed to random-effects model[70].

Next, MR-Egger test was performed. The MR-Egger test, in contrast to the inverse-variance-weighted method, does not assume all genetic variants to be valid[23]. The MR-Egger regressions allows for a variable intercept as a consequence of allowing genetic variants to be invalid. Large deviations from the non-zero intercept represent large average horizontal pleiotropic effect across the genetic variants[23]. An MR-Egger's intercept of zero, tested using a $P$ value threshold of

>0.05, was considered to provide evidence for absence of pleiotropic bias. The MR-Egger assumes that the association of genetic variants with the exposure are independent of the direct effects of the genetic variants on the outcome (InSIDE assumption)[23]. In case the InSIDE assumption is not violated, the slope coefficient from the MR-Egger regression is a consistent estimate of the causal effect. In addition, heterogeneity within the MR-Egger analysis was assessed by calculating Rucker's Q[70]. A significant difference ($P < 0.05$) between the Cochran's Q and Rucker's Q (Q-Q')[70] indicates the MR-Egger test to be a better method to study the genetic association between the particular exposure and outcome. MR-PRESSO was used to detect pleiotropy as well[71]. MR-PRESSO compares the difference between the residuals for each genetic variant in the variable, non-zero intercept of the genetic variant-outcome estimate with the genetic variant-exposure estimate in case pleiotropy is absent[71]. By doing so, pleiotropic effects can be detected and outliers can be identified. MR-PRESSO then re-analysis the association without the outliers, correcting for possible pleiotropic effects. Next, we excluded genetic variants with potential pleiotropic effects through education and through all traits (Supplementary Table 7), identified by query of the GWAS catalogue for genetic variants in LD > 0.8 with the newly identified variants for sedentary behaviours. Multivariable MR was used to understand the relationship between sedentary behaviours, education and CAD[65]. This allows for assessing the direct effect of sedentary behaviours on CAD, which is the effect of sedentary behaviours that is not driven by education. Previously published summary statistics of a GWAS performed on years of education in the SSGAC cohort was used as secondary exposure variable. Multivariable MR weighted regression methods were used, in which for each exposure the instruments are selected and regressed together against the outcome, weighting for the inverse variance of the outcome[65]. In addition, we evaluated $Q_{x1}$, $Q_{x2}$ and $Q_a$ in the two-sample multivariable MR. $Q_{x1}$ and $Q_{x2}$ provide information how much variance the genetic variants explain on the primary (sedentary behaviours) and secondary exposures (education). When both $Q_{x1}$ and $Q_{x2}$ are larger than the critical value for a the $\chi^2$ distribution, there is little evidence of weak instrument bias. We estimated the critical value for the $\chi^2$ distribution using the amount of SNPs minus two degrees of freedom at a $P$ value of 0.05. $Q_a$ was assessed to test for heterogeneity and thus potential pleiotropy in the multivariable two-sample MR setting. In case $Q_a$ is larger than the critical value on the $\chi^2$ distribution, there is evidence for heterogeneity and thus potential pleiotropy even when corrected for education. The critical value for the $\chi^2$ distribution was assessed using the amount of SNPs minus three degrees of freedom at a $P$ value of 0.05. For the results, please see Supplementary Table 11. Before multivariable MR analyses, we explored the association between years of education and sedentary behaviours using the inverse-variance-weighted fixed-effect method (Supplementary Data 11; Supplementary Tables 8–10). Multivariable MR has been proven to be a valid method for investigation of mediation as well[72]. Additional multivariable MR analyses were therefore performed to investigate the direct effect of sedentary behaviours on CAD, independent of traditional cardiovascular risk factors, and to explore potential mediation through these risk factors[72]. The traditional cardiovascular risk factors included BMI, history of diabetes, systolic blood pressure, diastolic blood pressure, history of hypertension and lipid profile. For this, all independent lead variants associated with sedentary behaviours at $P < 1 \times 10^{-8}$ were used. First, MR analyses between sedentary behaviours and the secondary phenotype were performed, as absence of such an association would suggest correction for the secondary exposure to be unnecessary. Next, a multivariable MR between sedentary behaviours, the secondary phenotype and CAD was performed. Beta's and standard errors for the secondary exposure were obtained within the UK Biobank using linear and logistic regression analyses. We note that the effects and standard errors of both exposures (sedentary behaviours and cardiovascular risk factors) are estimated in the same cohort of the UK Biobank and thus covariance in this setting could be reintroduced. However, as the effect of the secondary exposure can be overestimated when tested within the same cohort, this would likely results in a stricter correction within the multivariable MR. Lastly, MR-Steiger filtering was applied in the main analyses to remove variants that are more strongly associated with CAD than with sedentary behaviours[73]. MR-Steiger filtering calculates the $R^2$ for the exposure and outcome and removes variants if the $R^2$ of the exposure is significantly lower than $R^2$ of the outcome[73].

**Sensitivity analyses in MR analyses.** Several sensitivity analysis were performed. Weighted median analysis was performed, which allows up to 50% of information from variants to violate MR assumptions, in contrast to regular inverse-variance-weighted analysis in which absence of pleiotropic effects for all included genetic variants is assumed[74]. In addition, we performed weighted mode-based estimator MR analyses which allows even the majority of all variants to be invalid in case the largest number of that produce similar MR results are valid. Weighted mode-based MR generates causal effect estimates based on these valid instruments[75]. To investigate whether the results were robust to $P$ value thresholds, we determined the relationship between sedentary traits with CAD. This was done by adding genetic variants with higher $P$ values for sedentary behaviours ($P < 5 \times 10^{-8}$, $<1 \times 10^{-7}$, $<1 \times 10^{-6}$ and $<1 \times 10^{-5}$) and then repeated the MR analyses stated above. Lowering the $P$ value thresholds increases the chance of weak instrument bias and therefore instrument strength for all variants were assessed using the F-statistic (Supplementary Data 10).

**Reporting summary**. Further information on research design is available in the Nature Research Reporting Summary linked to this article.

**Data availability**

The data that support the findings of this study are available from the corresponding author upon reasonable request. The GWAS datasets summary statistics generated during the current study are available in the following repository, (https://doi.org/10.17632/mxjj6czsrd.1).

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

## Acknowledgements

This research has been conducted using the UK Biobank resource under the application number's 12006 and 15031. We thank the CARDIoGRAMplusC4D and SSGAC investigators for making their data publicly available. We would like to thank the Centre for Information Technology of the University of Groningen for their support and for providing access to the Peregrine high-performance computing cluster. In addition, we want to thank the "Medische en Informatie Technologie Systeembeheer" of the University Medical Center of Groningen for their support on and maintenance of our own computing cluster. We thank Ruben N. Eppinga, MD, PhD, Tom Hendriks, MD, M. Yldau van der Ende, MD, PhD, Yanick Hagemeijer, MSc, Hilde Groot, MD, and Jan-Walter Benjamins, BEng (Department of Cardiology, University of Groningen, University Medical Center Groningen, Groningen, the Netherlands), for their contributions to the extraction and processing of data in the UK Biobank. None of the contributors received compensation except for their employment at the University Medical Center Groningen. This work was also supported by NWO VENI (016.186.125) to N.V.

## Author contributions

N.V. conceived the idea for the study. P.v.d.H. provided supervision and obtained the genetic and observational data. N.V., Y.J.v.d.V. and M.A.S. were involved in data processing and statistical analyses. N.V., Y.J.v.d.V., P.v.d.H., M.R., M.A.S. were involved in interpreting the data. Y.J.v.d.V. wrote the first draft of the manuscript, and all authors critically revised it.

## Competing interests

The authors declare no competing interests.
