## [Peer Review File · Nature Communications]

Reviewers' Comments:

Reviewer #1:

Remarks to the Author:

The authors conducted MR experiments to examine the causal association between sedentary leisure behavior and CAD using 2-sample MR. They conducted GWAS for sedentary behavior to identify genome wide significant (GWS) variants and then tested these genetic instruments against CAD in an independent study of Cad genetics (CARDIoGRAMplusC4D). They showed a causal association between sedentary leisure behavior (television watching) and CAD. The studies are well conducted and well explained. They have comprehensively examined the associations both from a phenotypic and genetic standpoint. For the most part their results are clearly presented. They contextualize the study well and acknowledge its limitations. This is not the first study to address these questions. I have the following comments:

- 1) Please provide some quantitative results in the text lines 96-98.
- 2) The Manhattan plot is hard to interpret given it is for 3 traits. It should be reformatted or repalced.
- 3) Why were there very different findings with respect to the number of GWS variants for the 3 sedentary leisure time behavior traits? How much of the variance in each trait is explained by the genetics?
- 4) The authors provide the phenotyping correlation for each of the 3 sedentary leisure tme traits. They also identify a set of overlapping GWS variants. It would be interesting to compare how the genome wide genetic correlation (R_g) compares between the traits and to what degree this is similar to the phenotypic correlation.
- 5) Since the primary question is how sedentary leisure time behavior affects CAD, and the authors show a strong correlation between sedentary leisure time behavior and CAD, did they consider analyzing a composite sedentary leisure time behavior phenotype?
- 6) Physical activity should be strongly inversely correlated with sedentary behavior. It would be interesting to demonstrate an inverse causal association between physical activity and CAD as an internal control. Even though this might not be entirely novel, it would add to the paper.
- 7) The results from the multivariable MR are not given in the results section other than referring to the instrument on line 147. These need to be added to the results section.
- 8) It seems unlikely that television watching is directly causal on CAD, especially since the pathway analysis of the genetics of television watching implicate neurological pathways and not cardiovascular pathways. To what degree is the casual association between television watching and CAD mediated by known risk factors such as BMI, hypertension, lipids, and diabetes, all of which can result from a sedentary lifestyle. The paper and its novelty would be greatly enhanced by the use of mediation based MR to address this.
- 9) The authors allude to the availability and lack of availability of occupational data. To what degree might this be able to be brought to bear on the overall question of sedentary lifestyle and CAD?

Reviewer #2:

Remarks to the Author:

Sedentary behaviours and risk of CAD - GWAS and MR

van de Vegte et al

Comments to authors:

Overall this is an interesting application of the MR concept and analysis to the UKBB sample and to a complex exposure - sedentary time. The immediate thoughts when considering the subject of this GWAS and applied analysis are (i) the nature of the instrumentation and its impact on the applied analysis (MR) - i.e. the complexity of the behaviour being measured by genetic proxy and

the specificity of the observed signals to sedentary behaviour versus other things picked up and (ii) whether there is enough exploration the effects uncovered to merit the current analyses as a stand alone paper - i.e. that there is a two sample MR analysis in this paper, but no other exploration of the effects derived through alternative sources of evidence. The observational analysis within UKBB is of interest and gives a nice reappraisal of the relationship (association) between CAD events and activity - though it would have been nice to have a coincident analysis with the available actigraphy data. Indeed in the GWAS for the sedentary activity proxy measures (TV, driving, computer use), it would have been good to see the assessment of genetic overlap (rg) between these and other more objective measures of activity. There is an enrichment analysis - though the findings are general and reflect the nature of the GWAS undertaken (not sure how much can be taken from this at this stage).

Specific comments:

- It would have been great to cross-examine the use of these general measures for activity against the actigraphy data - through GWAS comparison and then in MR. Do these assess the same thing?
- It would be great to think about the overall predictive ability of the GWAS (all variants and not just those to be used as "instruments") in the assessment and or prediction of activity and then also CAD risk - would a score/predictor for sedentary behaviour perform well for disease?
- The performance of the "instruments" in MR ("instruments" in inverted commas as I have concerns that they are non-specific and may well predict other confounding factors) should be interrogated fully - what is the rg with other traits, are they independent of CAD risk factors - do they predict sedentary behaviour alone?
- Methodologically things are well undertaken - my only concern here would be that there is sufficient cross-examination of the product of the GWAS to allow proper interpretation of the MR analyses. Is it not clear that the effects described are specific to sedentary behaviour or just proxied well by it.
- Could more cross-examination of the bioinformatic analyses be provided - do the profiles generated look like that of other activity related GWAS traits?

Reviewer #3:

Remarks to the Author:

The overall objective of the manuscript by van de Vegte and colleagues was to identify genetic determinants of three distinct phenotypes of sedentary behavior and test whether or not the genetic determinants were casually related to coronary artery disease. This study is both innovative and significant as it clarifies biological mechanisms that provide context to findings already reported in observational studies showing individuals with higher time spent in sedentary behaviors have an increased risk of coronary artery disease, as well as premature mortality and other noncommunicable diseases. This paper will have a substantial impact on the field, and the authors should be commended for their thoughtful and thorough contribution to this growing body of evidence.

MAJOR CONCERNS:

None.

MODERATE CONCERNS:

1. Introduction and/or Discussion: To provide additional rationale on the significance of this work, authors should include language that the selected sedentary phenotypes (while obtained via report based methods) focus on behaviors that are largely modifiable via intervention, particularly those behaviors that are discretionary (e.g., television watching and non-occupational computer use).
2. Given the detail provided in the methods section on the item used to assess driving (see lines

270-271), this should not be labeled as "leisure driving" throughout the text. Driving can occur both during non-discretionary (e.g., commuting to work) and discretionary (e.g., drive to see the countryside) periods of the day, and not simply occur within the leisure (i.e., discretionary) periods of the day (see Pettee Gabriel, 2012, JPAH for a Conceptual Framework).

3. Discussion: It might be useful to add that television watching has long been used as a proxy for overall sedentary behaviors in observational studies. It's primarily used because, like leisure-time physical activity (i.e., sports/exercise), it is potentially modifiable and there is variability across study participants given it is an activity that individuals may choose to participate in (or not) during discretionary periods of the day that are not already designated for work and/or household/self-care/caretaking responsibilities. Further, measures that prompt individuals to recall and report total sedentary behaviors (sitting) are not particularly reliable or valid. This makes television viewing somewhat unique from reported computer use (e.g., individual may use the computer for leisurely pursuits, but also for more utilitarian activities such as paying bills) and driving. Thus, study findings showing the strongest genetic support for television watching and coronary artery disease are potentially even more significant because this particular phenotype is so well poised for intervention.

4. Discussion (lines 244-254): As noted by the authors, sedentary behaviors were based on participant report, and together did not fully characterize daily sedentary time. Given this, it would be useful for authors to provide areas of future research, including replication using accelerometer-based measures of sedentary behavior once sufficient sample size is available for analysis.

MINOR CONCERNS:

1. Line 71: "high levels" is more clearly revised to "prolonged time" given the way sedentary behaviors are often ascertained via reported methods.
2. Line 92: add "Mean daily reported leisure television watching...".
3. Line 140: add "...with television watching, compare to none...".
4. Line 256: add "... each 1.5 hour per day increase of sedentary behavior".
5. Line 294-296: The authors should be commended for adjusting for physical activity, however, the unit of expression for the physical activity data is unclear. Please provide: (1) detail on the threshold used to define "ideal cardiovascular health" and (2) examples of "do-it-yourself" physical activity types.
6. Table 1: Please include physical activity (and any other covariates that were included in the models, but not included in Table 1).

Point-by-point response to reviewers

We thank the reviewers for their comments on our manuscript and thank the editor for the chance to respond to them. We shall address the comments sequentially below.

Reviewer #1:

The authors conducted MR experiments to examine the causal association between sedentary leisure behavior and CAD using 2-sample MR. They conducted GWAS for sedentary behavior to identify genome wide significant (GWS) variants and then tested these genetic instruments against CAD in an independent study of Cad genetics (CARDIoGRAMplusC4D). They showed a causal association between sedentary leisure behavior (television watching) and CAD. The studies are well conducted and well explained. They have comprehensively examined the associations both from a phenotypic and genetic standpoint. For the most part their results are clearly presented. They contextualize the study well and acknowledge its limitations. This is not the first study to address these questions. I have the following comments:

1) Please provide some quantitative results in the text lines 96-98.

Response: We thank the reviewer for the overall positive assessment of the current work and are glad to hear the study is well conducted and explained. We agree that quantitative results should be reported and now included these in text lines 91-93.

2) The Manhattan plot is hard to interpret given it is for 3 traits. It should be reformatted or replaced.

Response: Thank you for addressing this point. We now included a new Manhattan plot with a different color scheme, which is easier interpret.

3) Why were there very different findings with respect to the number of GWAS variants for the 3 sedentary leisure time behavior traits? How much of the variance in each trait is explained by the genetics?

Response: We hypothesized and observed that these three sedentary behavior traits are different from each other. This is highlighted by different or opposing loci, opposing trait correlations and pathway analyses. Further, differences in heritability is the most likely explanation for the different findings with respect to the number of GWAS variants. SNP-based heritability was performed by BOLT-REML and shows large differences in heritability estimates as well (please see below). The following line was added to the method section:

“SNP-heritability estimates were assessed using BOLT-REML variance components analysis⁶.”

And the following line was added to the results section:

“SNP-heritability as estimated by BOLT-REML was highest for television watching ($h^2_g = 0.161$, $se = 0.002$), followed by leisure computer use ($h^2_g = 0.093$, $se = 0.002$), and driving ($h^2_g = 0.044$, $se = 0.002$).”

4) The authors provide the phenotyping correlation for each of the 3 sedentary leisure time traits. They also identify a set of overlapping GWAS variants. It would be interesting to compare how the genome wide genetic correlation (Rg) compares between the traits and to what degree this is similar to the phenotypic correlation.

Response: We performed additional genetic correlation analyses between the three sedentary phenotypes and added the following sentence to the Supplementary information:

“Genetic correlations between the three sedentary phenotypes were assessed using BOLT-REML variance components analysis⁶.”

The following results to the results section:

“Television watching and computer use showed a negative genetic correlation ($r_g = -0.281$, $se = 0.011$, $P = 6.17 \times 10^{-144}$), similar to the observational analyses. Driving was positively associated with television watching ($r_g = 0.231$, $se = 0.016$, $P = 3.00 \times 10^{-47}$), but not with computer use ($r_g = 0.013$, $se = 0.019$, $P = 0.494$).

And the following line in the discussion section:

“Observational and genetic correlations between sedentary behavior traits were weak and similar between both approaches for television watching and computer use.”

5) *Since the primary question is how sedentary leisure time behavior affects CAD, and the authors show a strong correlation between sedentary leisure time behavior and CAD, did they consider analyzing a composite sedentary leisure time behavior phenotype?*

Response: This was indeed considered, but not performed for several reasons. First and foremost, the three leisure sedentary behaviors showed little phenotypic and genetic overlap. Only 16 loci out of 169 were shared and out of those, nine had opposing effects. In addition, total sedentary behavior might be overestimated when sedentary behaviors are simply added together^{1,2}.

6) *Physical activity should be strongly inversely correlated with sedentary behavior. It would be interesting to demonstrate an inverse causal association between physical activity and CAD as an internal control. Even though this might not be entirely novel, it would add to the paper.*

Response: In the current version of the manuscript, we assessed the genetic correlation between sedentary behaviors and physical activity types (total, moderate and walking). As expected, these were inversely correlated with sedentary behaviors (please see **Supplementary Data 2**). However, it is beyond the scope of the current’s article research question to investigate the association between physical activity and CAD as well.

7) *The results from the multivariable MR are not given in the results section other than referring to the instrument on line 147. These need to be added to the results section.*

Response: We now included the following sentence in the results section:

“Using the multivariable MR approach, we found the direct effect of television watching on CAD to be attenuated to the total effect. However, a one SD increase in genetically determined leisure time watching television still increased the probability of CAD when corrected for education (OR 1.42, 95% CI 1.09-1.84, $P = 8.70 \times 10^{-03}$).”

8) *It seems unlikely that television watching is directly causal on CAD, especially since the pathway analysis of the genetics of television watching implicate neurological pathways and not cardiovascular pathways. To what degree is the casual association between television watching and CAD mediated by known risk factors such as BMI, hypertension, lipids, and diabetes, all of which can result from a sedentary lifestyle. The paper and its novelty would be greatly enhanced by the use of mediation based MR to address this.*

Response: We very much agree with the reviewer that other pathways could mediate the association sedentary behaviors and CAD. We therefore performed additional multivariable MR analyses

correcting for body mass index, history of diabetes, systolic blood pressure, diastolic blood pressure, history of hypertension and lipid profile. Details on the analyses were added to the Supplementary Information. The following sentence was added to the results section:

*“The direct effect of television watching on CAD was also attenuated compared to the total effect when corrected for BMI (OR 1.28, 95% CI 1.05-1.55, P= 0.01), low density lipid protein levels (OR 1.44, 95% CI 1.25-1.66, P= 6.2×10^{-07}), a history of diabetes (OR 1.31, 95% CI 1.11-1.55, P= 1.64×10^{-03}) and hypertension (OR 1.25, 95% CI 1.06-1.48, P= 8.2×10^{-03}). However, all associations remained significant. Please see **Supplementary Table 7** for the full results.”*

And the following sentence was added to the discussion section:

“We did observe vertical pleiotropy, as the association between television watching and CAD was attenuated when corrected for educational years and cardiovascular risk factors. This provides genetic insights in how complex traits as sedentary behaviors are associated with CAD. In the end, the effect of television watching on CAD remained significant throughout all multivariable MR analyses.”

9) The authors allude to the availability and lack of availability of occupational data. To what degree might this be able to be brought to bear on the overall question of sedentary lifestyle and CAD?

Response: The degree to which this might be able bring to bear on the overall question of sedentary lifestyle on CAD is that total sedentary behavior could have been assessed in case occupational data had been available, analogous to studies in which sedentary behavior is obtained through accelerometry measurement³. We changed the following sentence from:

“However, since the questionnaire only included leisure and not occupational sedentary behaviors, conclusions cannot be generalized to occupational sedentary behaviors.”

To:

“However, since the questionnaire did not include occupational sedentary behaviors, conclusions cannot be generalized to total sedentary behavior.”

Reviewer #2:

Overall this is an interesting application of the MR concept and analysis to the UKBB sample and to a complex exposure - sedentary time. The immediate thoughts when considering the subject of this GWAS and applied analysis are (i) the nature of the instrumentation and its impact on the applied analysis (MR) - i.e. the complexity of the behaviour being measured by genetic proxy and the specificity of the observed signals to sedentary behaviour versus other things picked up and (ii) whether there is enough exploration the effects uncovered to merit the current analyses as a stand alone paper - i.e. that there is a two sample MR analysis in this paper, but no other exploration of the effects derived through alternative sources of evidence. The observational analysis within UKBB is of interest and gives a nice reappraisal of the relationship (association) between CAD events and activity - though it would have been nice to have a coincident analysis with the available actigraphy data. Indeed in the GWAS for the sedentary activity proxy measures (TV, driving, computer use), it would have been good to see the assessment of genetic overlap (r_g) between these and other more objective measures of activity. There is an enrichment analysis - though the findings are general and reflect the nature of the GWAS undertaken (not sure how much can be taken from this at this stage).

1) It would have been great to cross-examine the use of these general measures for activity against the actigraphy data - through GWAS comparison and then in MR. Do these assess the same thing?

Response: We greatly appreciate the reviewer's interest in our paper. We agree with the reviewer that cross-examination is important and therefore performed genetic correlation analyses with the GWAS summary statistics of the accelerometer data from Doherty *et al.*⁴. We added the following sentence to the results section:

*“Television watching and computer use showed a positive genetic correlation with objectively measured sedentary behavior (respectively, $r_g = 0.145$, $se = 0.0284$, $P = 2.97 \times 10^{-7}$; $r_g = 0.4571$, $se = 0.03$, $p = 4.23 \times 10^{-52}$), while driving did not ($r_g = -0.029$, $se = 0.047$, $P = 0.535$). Genetic correlations with other traits can be found in **Supplementary Data 2.**”*

And the following sentence to the discussion section:

“The genetic correlation between television watching and objectively measured sedentary behavior was weak, in accordance to previous findings from observational studies²³. This is at least in part because accelerometers can only measure total sedentary time and not domain-specific behaviors, such as television watching²⁴. The correlation between computer use and objectively measured sedentary behavior was higher; one possible explanation for the current finding is a volunteering bias in the accelerometer data, which could have put forward individuals who were more highly educated and spent more leisure time on the computer.”

2) It would be great to think about the overall predictive ability of the GWAS (all variants and not just those to be used as “instruments”) in the assessment and or prediction of activity and then also CAD risk. Would a score/predictor for sedentary behaviour perform well for disease?

Response: The aim of the current article is to assess how genetics might affect sedentariness and to expand the current evidence on the association between sedentary behaviors and coronary artery disease. We also believe the GWAS should only be used as a tool to better understand biological and epidemiological pathways. A polygenic score will not have sufficient power to predict sedentariness in a population, in any meaningful way beyond what is presented by our Mendelian randomization study already.

3) *The performance of the “instruments” in MR (“instruments” in inverted commas as I have concerns that they are non-specific and may well predict other confounding factors) should be interrogated fully*

Response: We understand the reviewers concern on the specificity of the MR-instruments and this is why we had an extensive limitation section in the main manuscript and supplementary about this potential issue. In short, we performed rigorous sensitivity analyses; we excluded variants strongly associated ($R^2 > 0.8$) with any other trait. Besides multivariate Mendelian randomization analyses in which we corrected for educational years, we now added analyses in which we corrected for BMI, blood pressure, hypertension, lipid profile and diabetes (**Supplementary Table 7**). For further information, please see our response to question 7 from reviewer 1.

4) *What is the rg with other traits, are they independent of CAD risk factors - do they predict sedentary behaviour alone?*

Response: We performed additional genetic correlations analyses with other traits which can be found in **Supplementary Data 2**. The following sentences were added to the discussion section:

“We found the highest genetic correlations between sedentary behaviors and educational traits, which were negative for television watching and driving, and positive for computer use. In addition, sedentary behaviors as measured by leisure television watching and driving were correlated with obesity traits, but not waist-hip ratio. This is in accordance with the current understanding that overall fat distribution is mainly neurologically driven¹⁷, but waist-hip ratio by adipose pathways¹⁸. Intriguingly, computer use was not correlated with any obesity trait. Based on these results, we conclude that questionnaires have the ability to capture domain-specific aspects of sedentary behavior and share complex genetic patterns with intelligence-related traits and cardio-metabolic risk factors.”

Because of these high correlations, we performed additional multivariate Mendelian randomization analyses to correct for these traits and obtain the direct effect of sedentary behaviors on CAD. Please see our response to question 4, and question 7 from reviewer 1.

5) *Methodologically things are well undertaken - my only concern here would be that there is sufficient cross-examination of the product of the GWAS to allow proper interpretation of the MR analyses. Is it not clear that the effects described are specific to sedentary behaviour or just proxied well by it.*

Response: We thank the reviewer for their kind words and agree that there is not absolute certainty that the effects described are specific to sedentary behavior and therefore included the following sentence in the discussion section:

“Although weak instrument bias was carefully assessed, our analyses do not provide evidence for the specificity of the discovered genetic instruments for its phenotype. This is difficult to investigate, as there is a complex interplay between neurological driven traits, cardio-metabolic risk factors and disease. MR analyses should therefore be interpreted with caution, as these could be driven by related traits, i.e. pleiotropy.”

However, we provide thorough analyses of pleiotropy. Please also see the response to question 4 and question 7 from reviewer 1.

6) *Could more cross-examination of the bioinformatic analyses be provided - do the profiles generated look like that of other activity related GWAS traits?*

Response: Yes, the profiles of the bioinformatics analyses do match those of the previous activity GWAS's^{5,6}. The following sentence was added to the discussion section on lines 206-207:

"The important role of the central nervous system is analogous to earlier GWAS on physical activity questionnaires and device-measured activity data^{15,16}."

Reviewer #3:

The overall objective of the manuscript by van de Vegte and colleagues was to identify genetic determinants of three distinct phenotypes of sedentary behavior and test whether or not the genetic determinants were casually related to coronary artery disease. This study is both innovative and significant as it clarifies biological mechanisms that provide context to findings already reported in observational studies showing individuals with higher time spent in sedentary behaviors have an increased risk of coronary artery disease, as well as premature mortality and other noncommunicable diseases. This paper will have a substantial impact on the field, and the authors should be commended for their thoughtful and thorough contribution to this growing body of evidence.

1) Introduction and/or Discussion: To provide additional rationale on the significance of this work, authors should include language that the selected sedentary phenotypes (while obtained via report based methods) focus on behaviors that are largely modifiable via intervention, particularly those behaviors that are discretionary (e.g., television watching and non-occupational computer use).

Response: We thank the reviewer for the very kind words and are pleased to hear that he is confident about the impact this paper could have in the field of modifiable lifestyle risk behaviors. We agree with the reviewer that the modifiability of the lifestyle risk behaviors investigated in the article should be highlighted and therefore added the following sentence to the discussion section:

“The additional evidence on the association between television watching and CAD supports the rationale for interventions targeting television watching²⁸⁻³⁰, especially considering its high prevalence² and non-occupational characteristics.”

2) Given the detail provided in the methods section on the item used to assess driving (see lines 270-271), this should not be labeled as “leisure driving” throughout the text. Driving can occur both during non-discretionary (e.g., commuting to work) and discretionary (e.g., drive to see the countryside) periods of the day, and not simply occur within the leisure (i.e., discretionary) periods of the day (see Pettee Gabriel, 2012, JPAH for a Conceptual Framework).

Response: We thank the reviewer for spotting this error and providing this detailed summary. We changed “leisure driving” to “driving” throughout the manuscript.

3) Discussion: It might be useful to add that television watching has long been used as a proxy for overall sedentary behaviors in observational studies. It’s primarily used because, like leisure-time physical activity (i.e., sports/exercise), it is potentially modifiable and there is variability across study participants given it is an activity that individuals may choose to participate in (or not) during discretionary periods of the day that are not already designated for work and/or household/self-care/caretaking responsibilities. Further, measures that prompt individuals to recall and report total sedentary behaviors (sitting) are not particularly reliable or valid. This makes television viewing somewhat unique from reported computer use (e.g., individual may use the computer for leisurely pursuits, but also for more utilitarian activities such as paying bills) and driving. Thus, study findings showing the strongest genetic support for television watching and coronary artery disease are potentially even more significant because this particular phenotype is so well poised for intervention.

Response: We thank the reviewer for this important suggestion to highlight the importance of television watching as a sedentary trait. We included the following section in the discussion:

“Television watching is often used as proxy for total leisure sedentary behavior in observational studies, as television watching is almost solely performed in non-occupational setting, modifiable by intervention²⁸⁻³⁰ and shows higher validity than total sedentary behavior as it is easier to recall³¹.”

4) Discussion (lines 244-254): As noted by the authors, sedentary behaviors were based on participant report, and together did not fully characterize daily sedentary time. Given this, it would be useful for authors to provide areas of future research, including replication using accelerometer-based measures of sedentary behavior once sufficient sample size is available for analysis.

Response: We agree with the reviewer that describing future perspectives are an important point for discussion in this case. We included the following sentence in the discussion section:

“Future research efforts should be directed at expanding the current set of analyses to total sedentary behavior, physical activity behaviors and include accelerometer data, when new cohorts with sufficient genetic data become available”.

MINOR CONCERNS:

- 1) Line 71: “high levels” is more clearly revised to “prolonged time” given the way sedentary behaviors are often ascertained via reported methods.
- 2) Line 92: add “Mean daily reported leisure television watching...”.
- 3) Line 140: add “...with television watching, compare to none...”.
- 4) Line 256: add “... each 1.5 hour per day increase of sedentary behavior”.
- 5) Line 294-296: The authors should be commended for adjusting for physical activity, however, the unit of expression for the physical activity data is unclear. Please provide: (1) detail on the threshold used to define “ideal cardiovascular health” and (2) examples of “do-it-yourself” physical activity types.
- 6) Table 1: Please include physical activity (and any other covariates that were included in the models, but not included in Table 1).

Response: We appreciate the reviewer’s detailed assessment and for spotting these errors. We carefully addressed these minor concerns throughout the manuscript.

References

1. Prince, S. A., LeBlanc, A. G., Colley, R. C. & Saunders, T. J. Measurement of sedentary behaviour in population health surveys: a review and recommendations. *PeerJ* **5**, e4130 (2017).
2. Saunders, T. J., Prince, S. A. & Tremblay, M. S. Clustering of children's activity behaviour: the use of self-report versus direct measures. *Int. J. Behav. Nutr. Phys. Act.* **8**, 48 (2011).
3. Prince, S. A., LeBlanc, A. G., Colley, R. C. & Saunders, T. J. Measurement of sedentary behaviour in population health surveys: a review and recommendations. *PeerJ* **5**, e4130 (2017).
4. Doherty, A. *et al.* GWAS identifies 14 loci for device-measured physical activity and sleep duration. *Nat. Commun.* **9**, 5257 (2018).
5. Klimentidis, Y. C. *et al.* Genome-wide association study of habitual physical activity in over 377,000 UK Biobank participants identifies multiple variants including CADM2 and APOE. *Int. J. Obes.* **42**, 1161–1176 (2018).
6. Doherty, A. *et al.* GWAS identifies 10 loci for objectively-measured physical activity and sleep with causal roles in cardiometabolic disease. *bioRxiv* 261719 (2018). doi:10.1101/261719

Reviewers' Comments:

Reviewer #1:

Remarks to the Author:

Thank you for responding to my critiques. The manuscript is much improved.

Reviewer #2:

Remarks to the Author:

NCOMMS-19-23149A

Sedentary behaviors and risk of coronary artery disease: a genome wide analysis and Mendelian randomization

Comments to authors:

The authors have provided a series of clarifications to their submitted paper - thank you. These (in particular the genetic overlap analyses) have added considerably. However I still have some concerns over the interpretation of the results. The major problem for the analyses presented is the complexity of the genetic instruments for the exposure of interest. This has actually been confirmed in the analysis now provided which the authors refer to in the case of educational attainment, but also with the component parts of the sedentary behaviour proxy. The exposure is clearly complex and this combined with the power of the UKBB GWAS will lead to GWAS results reflecting the broad nature of the heritable contributions to these measures. It is this complexity that I was hoping to have cross-examined when I commented previously:

"The performance of the "instruments" in MR ("instruments" in inverted commas as I have concerns that they are non-specific and may well predict other confounding factors) should be interrogated fully ..."

One response to the challenge of clarifying the message in the analysis presented was to adjust for other factors:

"we now added analyses in which we corrected for BMI, blood pressure, hypertension, lipid profile and diabetes ..."

This is ok, however as per any other epidemiological analysis is clearly not a complete picture and is subject to the effectiveness of these measures in capturing the features which can confound the relationships targeted for causal analysis. It seems that this was done in the MR analysis too and whilst the logic of this seems sensible, the very fact that there were alterations in the effects before and after adjustments strengthens concerns that the "instruments" being used were themselves not unique to sedentary behaviour.

The comparison of genetic contributions to sedentary behaviour and educational attainment were very interesting and take this concern further by showing that these were the highest correlated of all traits tested. Of all genetic contributions difficult to interpret, educational attainment is likely one of the hardest and to then have a sedentary exposure instrument aligned to this introduces yet another layer of inferential challenge.

I had concern also over the assumption that the genetic overlap between obesity traits/whr with educational attainment naturally meant that there was a confirmation of neuro VS adiposity lines of causality and I think that more evidence is needed to support:

"...we conclude that questionnaires have the ability to capture domain-specific aspects of sedentary

behavior and share complex genetic patterns with intelligence-related traits and cardio-metabolic risk factors.”

Given these aspects, in the previous review I had noted that:

“it would be great to think about the overall predictive ability of the GWAS (all variants and not just those to be used as “instruments”) in the assessment and or prediction of activity and then also CAD risk. Would a score/predictor for sedentary behaviour perform well for disease?”

This was by no means suggesting that there should be a prediction model based approach to the use of the GWAS sedentary behaviour - rather that given the complexity of the “instrument” here, then one can feasibly abandon causality and in a non-casual framework look to assess the relationship between genetic variation in the entire genome and health outcomes. In this case (and as used elsewhere effectively - by the likes of Amit Khera and others), the complex (and complex) genetic measure of sedentary behaviour would be a tool for the measurement of this exposure in the absence of a better tool - the comparison of which to the current phenotyping would be an interesting analysis re. health outcomes.

Overall - this remains a really interesting paper and (bar the adjustments of MR analyses for covariables which is only interesting as a sensitivity analysis and not conclusive) the additional results add to the manuscript. However, the key iteration required is a clear contextualisation of the GWAS results and the MR analysis given the complex nature of the exposures being assessed. This may detract from the notion of sedentary behaviour being inceptive for CAD, however is a better reflection of the tests undertaken (arguably!).

Point-by-point response to reviewers

We thank the reviewer for the comments on our manuscript and want to thank the editor for the chance to respond to all points raised. We shall address the comments sequentially below.

Editor

Further, one reviewer mentioned that multivariable MR might be conflated with mediation-based MR in this study and as this might significantly influence the conclusions you can draw from these analyses we ask that you respond to this concern in your point-by-point response (before your responses to the reviewer comments).

Response: We thank the editor for bringing forward this important question from one of the reviewers. Although both multivariable MR and mediation-based MR (also referred to as two-step MR) have been proven to be valid approaches to investigate mediation¹, the methodologies are indeed different. Multivariable MR has several advantages over a two-step mediation-based MR¹, including the possibility to use overlapping instruments for multiple exposures¹ and to assess weak instrument bias and heterogeneity in the multivariable setting². We would like to assure that only multivariable Mendelian randomization analyses were performed in the current article.

We did not report on any two-step mediation-based MR and are therefore unsure how this might have been conflated with the multivariable MR analyses. We used the multivariable MR analyses to (1.) provide evidence whether secondary exposures like cardiovascular risk factors might be on the causal pathway between sedentary behavior and CAD (in response an earlier comment of reviewer #2) and (2.) to estimate the direct effect of sedentary behaviors on CAD analogous to the observational analyses, in light of overlap with educational attainment. We do not consider this to be a mediation analysis per se, and have not described it as such in the manuscript. We have further elaborated on the limitations of multivariable MR analyses, also in response to the reviewer below.

The reviewer might have misread that we performed a two-step MR because we did perform a MR between the primary and secondary exposures (analogous to the first step performed in two-step MR)³. We performed this as a first step in the multivariable MR as lack of an association would suggest correction for the secondary exposure to be unnecessary, since total and direct effect would likely be equal².

Reviewer #2:

The authors have provided a series of clarifications to their submitted paper - thank you. These (in particular the genetic overlap analyses) have added considerably. However I still have some concerns over the interpretation of the results. The major problem for the analyses presented is the complexity of the genetic instruments for the exposure of interest. This has actually been confirmed in the analysis now provided which the authors refer to in the case of educational attainment, but also with the component parts of the sedentary behaviour proxy. The exposure is clearly complex and this combined with the power of the UKBB GWAS will lead to GWAS results reflecting the broad nature of the heritable contributions to these measures. It is this complexity that I was hoping to have cross-examined when I commented previously: "The performance of the "instruments" in MR ("instruments" in inverted commas as I have concerns that they are non-specific and may well predict other confounding factors) should be interrogated fully ..."

Response: We thank the reviewer for the overall positive assessment of the revisions and are glad to hear the new analyses add to the current study in a substantial amount.

One response to the challenge of clarifying the message in the analysis presented was to adjust for other factors: “we now added analyses in which we corrected for BMI, blood pressure, hypertension, lipid profile and diabetes ...”

This is ok, however as per any other epidemiological analysis is clearly not a complete picture and is subject to the effectiveness of these measures in capturing the features which can confound the relationships targeted for causal analysis. It seems that this was done in the MR analysis too and whilst the logic of this seems sensible, the very fact that there were alterations in the effects before and after adjustments strengthens concerns that the “instruments” being used were themselves not unique to sedentary behaviour.

Response: We agree with the reviewer that every measures’ effectiveness strongly depends on how well it capture its’ features. Indeed, a relatively “crude”, but nonetheless valid⁴, measure as a questionnaire would therefore result in possible measurement error on the exposure. However, Mendelian randomization is a form of instrumental variable analysis. This type of analysis is known to be less likely affected by measurement error on the exposures (in this case, both primary and secondary in multivariable Mendelian randomization) than conventional analyses⁵.

We have added the following sentence in the limitations regarding these points:

“Both the observational and MR study are limited by quality of the questionnaires and the effectiveness of measurements to capture features that are on the causal pathways. However, MR studies are less likely to be affected by measurement error on the exposures than conventional observational analyses⁵.”

We agree with the reviewer that we cannot make claims on the ‘uniqueness’ or specificity of the instruments used for sedentary behavior. This is a disadvantage of taking a statistical approach (instruments associated with the exposure of interest at a given level of statistical significance), rather than a biological approach (taking forward instruments in a gene known to be associated with a trait). Current MR results could therefore be biased by pleiotropy and to address this we performed many sensitivity analyses in the manuscript according to the latest guidelines⁶, in which we payed extra attention to the complicated relationship with education⁷.

Nonetheless, our interpretation should be more nuanced as bias through pleiotropy is still impossible to rule out. We included a **Supplementary Discussion** in which we extensively debate current MR results, revised the discussion and added the following sentence in the discussion section:

“In addition, the genetic instruments could be non-specific to sedentary behaviors as a statistical and not a biological approach was used for their selection. Furthermore, we found evidence for heterogeneity and thus potential pleiotropy in the multivariable MR corrected for education, suggesting that unobserved confounders could play a role in the relationship between television watching and CAD. As far as verifiable using currently available methods⁶, all results point to the same direction and therefore seem to support the rationale that interventions targeting television watching may reduce CAD risk⁸⁻¹⁰, especially considering the high prevalence¹¹ and non-occupational characteristics of television watching.

The comparison of genetic contributions to sedentary behaviour and educational attainment were very interesting and take this concern further by showing that these were the highest correlated of all traits tested. Of all genetic contributions difficult to interpret, educational attainment is likely one of the hardest and to then have a sedentary exposure instrument aligned to this introduces yet another layer of inferential challenge.

Response: Thank you for addressing this point. This is indeed true and the very reason as to why we included the many sensitivity analyses in the manuscript to address this. We did not solely perform a multivariate Mendelian randomization, but we also excluded variants highly correlated with

educational traits to indicate that the observed effect is independent from education because this has been a strong confounder traditionally. We have now included this as a limitation:

*“To address the assumptions of MR and control for different types of biases, we performed several sensitivity analyses according to the latest guidelines⁶, each with their own strengths and weaknesses that are more extensively described in the **Supplementary Methods and Discussion**. It is important to recognize these limitations and strengths in light of the potential complicated relationship between education and disease⁷.”*

We also clarified the following section in the methods to elaborate on our choice to do a multivariable MR using education as secondary exposure:

“We paid extra attention to potential pleiotropic effects due to education, as this is the most often investigated determinant of sedentary behaviors¹² and a well-established predictor of CAD¹³. Multivariable MR analyses using education as a secondary exposure were performed as sensitivity analyses to assess independence of education in our estimates².”

In addition, we performed additional analyses to investigate whether the assumptions of the multivariable MR were fulfilled. In this two-sample multivariable MR setting, we evaluated weak-instrument bias using Q_{x1} and Q_{x2} , and heterogeneity and thus potential pleiotropy using Q_a , as further described in the **Supplementary Methods and Discussion**². In short, we found no evidence of weak instrument bias in the estimates, but Q_a indicated remaining heterogeneity and thus potential pleiotropy in the estimates between television watching, education and CAD.

I had concern also over the assumption that the genetic overlap between obesity traits/whr with educational attainment naturally meant that there was a confirmation of neuro VS adiposity lines of causality and I think that more evidence is needed to support: “...we conclude that questionnaires have the ability to capture domain-specific aspects of sedentary behavior and share complex genetic patterns with intelligence-related traits and cardio-metabolic risk factors.”

Response: Thank you for addressing this point. Previous studies on physical activity and sedentary behaviors found similar high correlations with education or intelligence-related traits and cardio-metabolic risk factors^{14,15}. We agree the current formulation might be overstated. We changed the sentence to:

“This suggests that questionnaires have the ability to capture domain-specific aspects of sedentary behavior as previously described⁴ and possibly share complex genetic patterns with education-related traits and cardio-metabolic risk factors.”

Given these aspects, in the previous review I had noted that:

“it would be great to think about the overall predictive ability of the GWAS (all variants and not just those to be used as “instruments”) in the assessment and or prediction of activity and then also CAD risk. Would a score/predictor for sedentary behaviour perform well for disease?”

This was by no means suggesting that there should be a prediction model based approach to the use of the GWAS sedentary behaviour - rather that given the complexity of the “instrument” here, then one can feasibly abandon causality and in a non-casual framework look to assess the relationship between genetic variation in the entire genome and health outcomes. In this case (and as used elsewhere effectively - by the likes of Amit Khera and others), the complex (and complex) genetic measure of sedentary behaviour would be a tool for the measurement of this exposure in the absence of a better tool - the comparison of which to the current phenotyping would be an interesting analysis re. health outcomes.

Response: Thank you for the clarifying the previous question. Indeed, the studies of for example Khera *et al.* investigated the predictive ability of commonly studied complex traits as coronary artery disease and body mass index^{16,17}. Although we believe this to be an interesting analysis but that it is beyond the scope of the current paper and that it would be more suitable for follow-up research. We also think that such an analysis will have many similar limitations as the current two-sample Mendelian randomization analysis but without the ability to perform extensive sensitivity analyses on the individual variant level, and therefore may not provide much additional insights.

Moreover, it is difficult at this point to carry out such an investigation confidently because of a lack of independent cohorts with similar sedentary behavior questionnaires and sufficient sample sizes for replication. We listed this as an issue in the limitations in the manuscript:

“Currently, the ability to replicate genetic variants in external cohorts is limited due a lack of available data concerning the same sedentary behavior questions and genetics.”

We also added the heritability estimates to provide insights in the predictive ability of the GWAS’s in the UK Biobank itself.

Overall - this remains a really interesting paper and (bar the adjustments of MR analyses for covariables which is only interesting as a sensitivity analysis and not conclusive) the additional results add to the manuscript. However, the key iteration required is a clear contextualisation of the GWAS results and the MR analysis given the complex nature of the exposures being assessed. This may detract from the notion of sedentary behaviour being inceptive for CAD, however is a better reflection of the tests undertaken (arguably!).

Response: Again, we thank the reviewer for the positive assessment of the revisions and are glad to hear the new analyses added to the current study. We hope the changes in the current version of the manuscript add to the interpretation of the results.

References

1. Carter, A. R. *et al.* Mendelian randomisation for mediation analysis: current methods and challenges for implementation. *bioRxiv* 835819 (2019). doi:10.1101/835819
2. Sanderson, E., Davey Smith, G., Windmeijer, F. & Bowden, J. An examination of multivariable Mendelian randomization in the single-sample and two-sample summary data settings. *Int. J. Epidemiol.* (2018). doi:10.1093/ije/dyy262
3. Relton, C. L. & Davey Smith, G. Two-step epigenetic mendelian randomization: A strategy for establishing the causal role of epigenetic processes in pathways to disease. *Int. J. Epidemiol.* **41**, 161–176 (2012).
4. Prince, S. A., LeBlanc, A. G., Colley, R. C. & Saunders, T. J. Measurement of sedentary behaviour in population health surveys: a review and recommendations. *PeerJ* **5**, e4130 (2017).
5. Sargan, J. D. The Estimation of Economic Relationships using Instrumental Variables. *Econometrica* **26**, 393 (1958).
6. Burgess, S. *et al.* Guidelines for performing Mendelian randomization investigations. *Wellcome Open Res.* **4**, 186 (2019).
7. Davies, N. M. *et al.* Multivariable two-sample Mendelian randomization estimates of the effects of intelligence and education on health. *Elife* **8**, (2019).
8. Otten, J. J., Jones, K. E., Littenberg, B. & Harvey-Berino, J. Effects of Television Viewing Reduction on Energy Intake and Expenditure in Overweight and Obese Adults. *Arch. Intern. Med.* **169**, 2109 (2009).
9. Raynor, H. A. *et al.* Reducing TV Watching During Adult Obesity Treatment: Two Pilot Randomized Controlled Trials. *Behav. Ther.* **44**, 674–685 (2013).
10. Keadle, S. K., Arem, H., Moore, S. C., Sampson, J. N. & Matthews, C. E. Impact of changes in television viewing time and physical activity on longevity: a prospective cohort study. *Int. J. Behav. Nutr. Phys. Act.* **12**, 156 (2015).
11. British heart foundation. BHF Cardiovascular Disease Statistics - UK Factsheet. (2017). Available at: <https://webcache.googleusercontent.com/search?q=cache:lRvsOXR1RDUI:https://www.bhf.org.uk/-/media/files/research/heart-statistics/physical-inactivity-report---mymarathon-final.pdf+%&cd=2&hl=en&ct=clnk&gl=nl>. (Accessed: 12th December 2018)
12. Prince, S. A., Reed, J. L., McFetridge, C., Tremblay, M. S. & Reid, R. D. Correlates of sedentary behaviour in adults: a systematic review. *Obes. Rev.* **18**, 915–935 (2017).
13. Tillmann, T. *et al.* Education and coronary heart disease: mendelian randomisation study. *BMJ* **358**, j3542 (2017).
14. Klimentidis, Y. C. *et al.* Genome-wide association study of habitual physical activity in over 377,000 UK Biobank participants identifies multiple variants including CADM2 and APOE. *Int. J. Obes.* **42**, 1161–1176 (2018).
15. Doherty, A. *et al.* GWAS identifies 10 loci for objectively-measured physical activity and sleep with causal roles in cardiometabolic disease. *bioRxiv* 261719 (2018). doi:10.1101/261719
16. Khera, A. V. *et al.* Genome-wide polygenic scores for common diseases identify individuals with risk equivalent to monogenic mutations. *Nat. Genet.* **50**, 1219–1224 (2018).
17. Khera, A. V. *et al.* Polygenic Prediction of Weight and Obesity Trajectories from Birth to Adulthood. *Cell* **177**, 587-596.e9 (2019).

Reviewers' Comments:

Reviewer #2:

Remarks to the Author:

Sedentary behaviors and risk of coronary artery disease: a genome wide analysis and Mendelian randomization.

van de Vegte et al

Comments to authors:

Reading the new version of this paper there were a series of broad points which remain or are now requiring attention:

(i) The introduction is relatively limited and does not consider the strengths and or problems with measuring and analysing activity and outcomes in an observational context. I think that the MR paragraph could likely go in place of this. Furthermore the last paragraph of the instruction should really not include main results.

(ii) In the early sections of the observational results, it would also be great to see the relationships between activity traits and outcomes and potential confounding factors.

(iii) In the main text it is not completely clear where the conditions of UKBB GWAS are (cleaning of data, prep and running of analysis) are reported?

(iv) It is unclear in this analysis (as this is a 2SMR) what value the F-statistic is in the MR analysis. The notion of F (as in a TLS analysis for one sample MR) is less relevant in the context of 2SMR and this derived metric is very closely associated with variance explained in the GWAS. It may be better to just report on the variance explained in exposure of the instruments in question.

(v) It is important to show all variants first before removing those flagged by MR-PRESSO. Ideally the latter analysis would remain a sensitivity with the main result being the scatter plot of instrument effects on risk factor and outcome (and related MR estimates).

(vi) It is not entirely clear that the informatics sections (or the strengths section) of the paper add a great deal. Arguably more could go into the introduction to set the problem and advantages of this work. the performance and nature of the instruments developed and potentially the role of LDSR in this paper (with exposures, outcomes and perceived confounders). These extensions would help the interpretation of the eventual MR results. I am not sure that the entry below gets around this:

"We advocate that the data presented here should be re-analyzed when MR guidelines are updated and MR methods further developed"

(vi) MR does not estimate "genetically determined" effects, rather the causal effect of exposure on outcome.

Minor comments:

(i) MR analyses do not "indicate" that a risk factor causes something else/risk, rather they provide estimates of the causal effects (given the relative strengths and limitations of the analysis. Care is needed with the interpretation and their working throughout

(ii) With the heritability estimates as they are, could a comment as to the power of the rg LDSR

analysis be made?

(iii) Please remove "significant" from statements about observed effects (e.g. the pathway analysis paragraph)

(iv) The section entitled "The genetic association between sedentary behaviors and CAD" should really be changes to something like "Estimating the causal relationship between sedentary behaviour and CAD"

Returning to the previous comments made:

It was good to see that some of these have been addressed, or at least have had a justified omission. In light of comments above and the previous comments made, however, some aspects appear to be still unaddressed. Given the status of the paper now, it may be appropriate to allow for editorial oversight re. the final version of the paper - however this will be subject to the editor reviewing these comments.

(i) A point not really addressed since the last review:

"The major problem for the analyses presented is the complexity of the genetic instruments for the exposure of interest. This has actually been confirmed in the analysis now provided which the authors refer to in the case of educational attainment, but also with the component parts of the sedentary behaviour proxy. The exposure is clearly complex and this combined with the power of the UKBB GWAS will lead to GWAS results reflecting the broad nature of the heritable contributions to these measures. It is this complexity that I was hoping to have cross-examined when I commented previously: "The performance of the "instruments" in MR ("instruments" in inverted commas as I have concerns that they are non-specific and may well predict other confounding factors) should be interrogated fully ..."

This is really referring to an exploration of the properties of the instruments being deployed in these analyses - for example, other traits which are explained by them or possible shared heritable contributions with other complex exposures.

Indeed there is now a section in the latter stages of the paper that reads:

"To overcome confounding, we used a MR approach..."

This is one of the potential benefits of MR, but potentially not where complex phenotypes have been entered into the GWAS for exposure and are likely (given the power of UKBB GWAS analysis) to yield complex instruments.

(ii) In response to the concerns over adjustments to clarify analyses:

"We have added the following sentence in the limitations regarding these points:

"Both the observational and MR study are limited by quality of the questionnaires and the effectiveness of measurements to capture features that are on the causal pathways. However, MR studies are less likely to be affected by measurement error on the exposures..."

This is ok, but the problem in this instance is the lack of clarity (likely) in the instruments - which themselves may well remain confounded. It would be great if this could be acknowledged directly.

(iii) The additional section to the discussion (pertinent to the limitations of the MR used here) are

well received.

(iv) MR sensitivity analyses are good (though don't address the above completely). To this end the addition below is good:

"we performed additional analyses to investigate whether the assumptions of the multivariable MR were fulfilled. In this two-sample multivariable MR setting, we evaluated weak- instrument bias using Qx1 and Qx2, and heterogeneity and thus potential pleiotropy using Qa, as further described in the Supplementary Methods and Discussion . In short, we found no evidence of weak instrument bias in the estimates, but Qa indicated remaining heterogeneity and thus potential pleiotropy in the estimates between television watching, education and CAD."

I tend to think, however, that a full exploration of the instruments in question would be best to feature in the main text.

(v) I still feel that the deployment of a genome wide predictor in this context would be both useful as a comparator, but also potentially in light of possible clinical application in this area. It is a shame that this is not considered to be in the remit of the study.

Point-by-point response to the reviewer

We thank the reviewer for the comments on our manuscript and we shall address the comments sequentially below.

(i) The introduction is relatively limited and does not consider the strengths and or problems with measuring and analysing activity and outcomes in an observational context. I think that the MR paragraph could likely go in place of this. Furthermore the last paragraph of the instruction should really not include main results.

Response: We agree with the reviewer on this subject. We now extended the introduction section significantly with the strengths and problems when analyzing sedentary behaviors in an observational context. However, we still summarize the major results and conclusions at the end of the introduction, as this is one of *Nature Communications*' style requirements.

(ii) In the early sections of the observational results, it would also be great to see the relationships between activity traits and outcomes and potential confounding factors.

Response: We now included the association with potential confounders in the current manuscript. The following sentence was included in the methods section:

“Associations between sedentary behaviours and potential confounders were assessed using linear regressions analyses or logistic regression analyses in case of binary outcomes.”

These associations are in general in the same direction as the genetically estimated correlations. For the full results please see **Supplementary Table 2**.

(iii) In the main text it is not completely clear where the conditions of UKBB GWAS are (cleaning of data, prep and running of analysis) are reported?

Response: We agree with the reviewer that this is not clear in the main text. We moved the information of the conditions of the GWAS from the supplement to the Methods section of the main manuscript.

(iv) It is unclear in this analysis (as this is a 2SMR) what value the F-statistic is in the MR analysis. The notion of F (as in a TSLS analysis for one sample MR) is less relevant in the context of 2SMR and this derived metric is very closely associated with variance explained in the GWAS. It may be better to just report on the variance explained in exposure of the instruments in question.

Response: Weak instrument bias is indeed a larger issue when using an one sample MR approach¹ and is by definition closely related to the variance explained by the instrument². However, performing F-statistic is currently recommended in both the one and two sample MR approach^{1,3}. Nonetheless, we do understand the importance of reporting the variance explained by the instruments and therefore report this in **Supplementary Data 10**.

(v) It is important to show all variants first before removing those flagged by MR-PRESSO. Ideally the latter analysis would remain a sensitivity with the main result being the scatter plot of instrument effects on risk factor and outcome (and related MR estimates).

Response: We revised the order of the results in the current version of the manuscript as suggested. We hope this improves the flow and readability of the section. We now also included the scatter plots of the main results in **Supplementary Figure 4-6**.

(vi) It is not entirely clear that the informatics sections (or the strengths section) of the paper add a great deal. Arguably more could go into the introduction to set the problem and advantages of this work, the performance and nature of the instruments developed and potentially the role of LDSR in this paper (with exposures, outcomes and perceived confounders). These extensions would help the interpretation of the eventual MR results. I am not sure that the entry below gets around this:

“We advocate that the data presented here should be re-analyzed when MR guidelines are updated and MR methods further developed”

Response: We brought forward the issue of the performance and nature of the instruments developed in the limitation section and now acknowledge this directly. In addition, we added the following sentence to the introduction section to bring forward the complex nature of sedentary behaviors and their genetic correlation with other traits:

“Considering the complex nature of behavioral traits and the broad range of determinants known to affect sedentary behaviors, we estimate the genetic correlation with other traits and find especially high correlations with educational and obesity traits.”

To better get around the main issue, i.e. possible pleiotropy in the analyses caused by the broad nature of the instruments, we changed the following sentence:

“We advocate that the data presented here should be re-analyzed when MR guidelines are updated and MR methods further developed”

To:

“We advocate that the data presented here should be re-analysed when MR methods to account for pleiotropy are further developed”

This does not get around all possible issues in the current MR analyses, but we therefore refer to the **Supplementary Discussion**, in which problems and advantages of this work in the context of MR are more extensively described.

(vi) MR does not estimate “genetically determined” effects, rather the causal effect of exposure on outcome.

Response: We changed “genetically determined effects” to the suggested “estimation of causal effects” within the MR context throughout the manuscript and supplementary information.

Minor comments:

(i) MR analyses do not “indicate” that a risk factor causes something else/risk, rather they provide estimates of the causal effects (given the relative strengths and limitations of the analysis. Care is needed with the interpretation and their working throughout

Response: We thank the reviewer for pointing this out and changed all sentences containing the previous phrasing to the correct phrasing.

(ii) With the heritability estimates as they are, could a comment as to the power of the rg LDSR analysis be made?

Response: LD Score regression has been proven to provide reasonably robust estimates of the genetic correlation⁴, but has been shown to be less accurate than for example Genomic Restricted Maximum Likelihood method under certain conditions⁵. However, we believe the current estimates to be accurate

since 1) current effect sizes were estimated within a single cohort of the UK Biobank, 2) most LD scores have been calculated using outcomes measured in the same sample of the UK Biobank (496 out of 696 traits) and 3) we included a high amount of variants ($N_{\text{variants}} = 19,400,418$). Under these conditions, LD Score regression has been shown to be approximately as accurate as the Genomic Restricted Maximum Likelihood method⁵.

(iii) Please remove “significant” from statements about observed effects (e.g. the pathway analysis paragraph)

Response: We removed all statements on significance from the pathway analyses paragraph.

(iv) The section entitled “The genetic association between sedentary behaviors and CAD” should really be changes to something like “Estimating the causal relationship between sedentary behaviour and CAD”

Response: Considering the fact we were limited to 60 character titles, we changed the header to:

“The causal relationship between sedentary behaviour and CAD”

We made sure to state the fact that it MR will always be an estimation of the causal effect in the first sentence of this section.

References

1. Burgess, S. *et al.* Guidelines for performing Mendelian randomization investigations. *Wellcome Open Res.* **4**, 186 (2019).
2. Palmer, T. M. *et al.* Using multiple genetic variants as instrumental variables for modifiable risk factors. *Stat. Methods Med. Res.* **21**, 223–242
3. Lawlor, D. A. Commentary: Two-sample Mendelian randomization: opportunities and challenges. *Int. J. Epidemiol.* **45**, 908–915 (2016).
4. Lee, J. J., McGue, M., Iacono, W. G. & Chow, C. C. The accuracy of LD Score regression as an estimator of confounding and genetic correlations in genome-wide association studies. *Genet. Epidemiol.* **42**, 783–795 (2018).
5. Ni, G. *et al.* Estimation of Genetic Correlation via Linkage Disequilibrium Score Regression and Genomic Restricted Maximum Likelihood. *Am. J. Hum. Genet.* **102**, 1185–1194 (2018).